# WebSeer: Training Deeper Search Agents through Reinforcement Learning with Self-Reflection

**Guanzhong He, Zhen Yang, Jinxin Liu, Bin Xu[†], Lei Hou, Juanzi Li**
Department of Computer Science and Technology, Tsinghua University
`he-gz24@mails.tsinghua.edu.cn`
`{xubin, lijuanzi, houlei}@tsinghua.edu.cn`

## Abstract

Search agents have achieved significant advancements in enabling intelligent information retrieval and decision-making within interactive environments. Although reinforcement learning has been employed to train agentic models capable of more dynamic interactive retrieval, existing methods are limited by shallow tool-use depth and the accumulation of errors over multiple iterative interactions. In this paper, we present WebSeer, a more intelligent search agent trained via reinforcement learning enhanced with a self-reflection mechanism. Specifically, we construct a large dataset annotated with reflection patterns and design a two-stage training framework that unifies cold start and reinforcement learning within the self-reflection paradigm for real-world web-based environments, which enables the model to generate longer and more reflective tool-use trajectories. Our approach substantially extends tool-use chains and improves answer accuracy. Using a single 14B model, we achieve state-of-the-art results on HotpotQA and SimpleQA, with accuracies of 72.3% and 90.0%, respectively, and demonstrate strong generalization to out-of-distribution datasets. The code is available at Github/WebSeer.

## 1 Introduction

Large language models (LLMs) have demonstrated remarkable performance across a wide range of natural language processing tasks, including question answering, summarization, and dialogue generation (Hendrycks et al., 2021; Rein et al., 2024). However, relying solely on the parametric knowledge of language models poses fundamental limitations: it is static, often outdated, and prone to hallucinations (Sardana, 2025). To overcome these challenges, retrieval-augmented generation (RAG) (Lewis et al., 2020) approaches have been developed to enable models to access and retrieve external documents dynamically.

With the continuous advancement of model capabilities, agentic RAG (Trivedi et al., 2023; Li et al., 2025; Jin et al., 2025) has emerged as a powerful paradigm. This design empowers models to follow more complex reasoning trajectories. Unlike traditional RAG systems (Asai et al., 2023; Trivedi et al., 2023; Yu et al., 2024), agentic RAG can freely browse vast knowledge sources available on the internet and leverage tools such as code execution to extend their skills, enabling them to tackle a more diverse range of tasks. While agentic RAG greatly extends the scope of tool use and demonstrates strong potential, existing approaches exhibits several notable limitations in practice. In particular, when faced with complex or open-domain tasks, current systems often struggle to maintain coherent reasoning chains and robust retrieval. Errors introduced at intermediate steps can easily accumulate, while the lack of effective coordination across components makes it difficult to achieve reliable end-to-end performance. Thus, despite its promise, agentic RAG still confronts a set of fundamental challenges that must be addressed.

**1) Insufficient Search Calls.** The most common issue when it comes to model invocation tools to solve problems is Insufficient API Calls (Kokane et al., 2025), which is also often seen in the RAG scenario, as shown in the Appendix B. This may be due to models being biased toward synthesizing the currently available information into a plausible answer rather than actively seeking

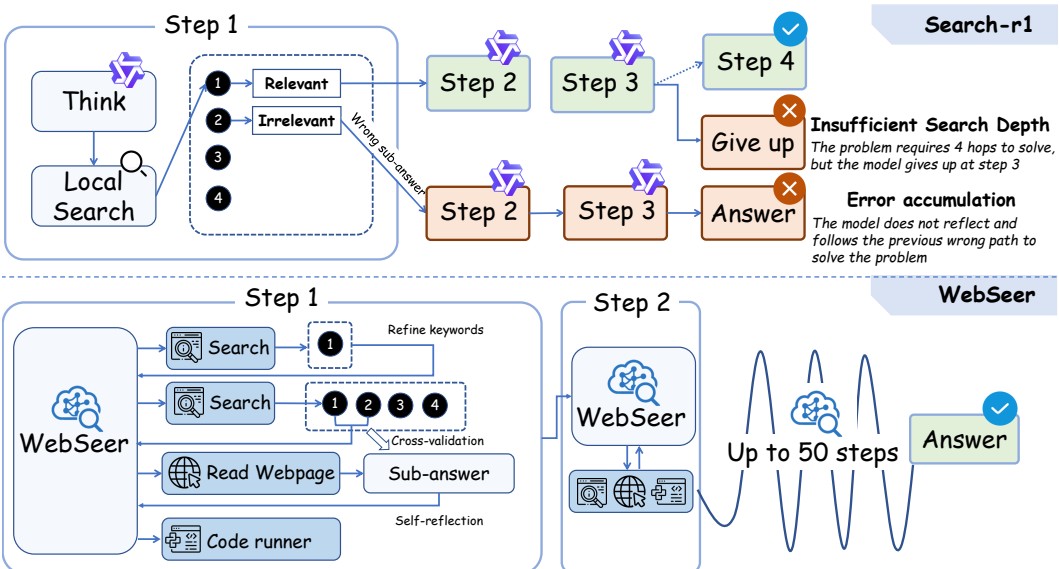

Figure 1: Comparasion between different search agent: Search-r1(Jin et al., 2025) faces shallow search and error buildup, while WebSeer trained using our self-reflection paradigm significantly improve tool calls and maintain rigorous reasoning.

new or complementary knowledge. Existing work typically exhibits short tool-use chains (Jin et al., 2025; Song et al., 2025), suggesting that models may fall into suboptimal behaviors—prematurely producing an answer instead of verifying its correctness or exploring further evidence.

**2) Lack of Spontaneous Self-Reflection Mechanisms.** Current search agents lack spontaneous reflection steps in RAG scenario: models neither actively cross-verify information nor autonomously rewrite queries or backtrack retrievals when uncertain. As a result, if initial retrieval is incomplete, the generation phase expands answers based on flawed or partial context, amplifying early errors.

**3) Neglect of Real-World Web Scenarios.** Most existing work focuses on retrieval from local vector databases, with limited attention to more complex and open-ended web agent scenarios.

In this paper, we introduce Webseer, a novel search agent designed to tackle complex real-world multi-hop question answering tasks. Different from prior approaches (Jin et al., 2025; Zheng et al., 2025), WebSeer explicitly encourages deeper exploration and integrates a build-in self-reflection mechanism, enabling the model to backtrack, reformulate queries, and iteratively refine its reasoning process. As shown in Figure 1, previous agentic RAG primarily rely on short tool-use chains and often terminate once a superficially plausible answer is formed. In contrast, WebSeer actively prolongs the search trajectory and incorporates reflection steps, and revises its queries when uncertainty is detected. This design enables Webseer to gather more comprehensive evidence, mitigate the accumulation of errors, and improve robustness in open-domian multi-hop reasoning. Specifically, we design a two-stage training framework that unifies cold start and reinforcement learning within the self-reflection paradigm. A central component of this framework is Self-Reflective Reinforcement Learning (SRRL), which leverages answer correctness signals during multi-turn interactions to more effectively encourage reflective behavior. To ground the model in realistic web scenarios, we equip it with three complementary tools: a web search API for external knowledge acquisition, a webpage reader for lightweight comprehension of web content, and a code executor for precise computation.

To support this framework, we construct a high-quality dataset of long-horizon reasoning trajectories through rejection sampling. These trajectories contain multiple rounds of answer refinement and substantially longer tool-use chains compared to conventional dialogue datasets. This training framework significantly increases the average length of tool invocation chains while maintaining rigorous reasoning quality. Compared to prior work (Zheng et al., 2025), all decisions and tool interactions are handled by a single model, eliminating the need for auxiliary agent controllers or stronger backbone models.

Overall, our contributions are three-fold:

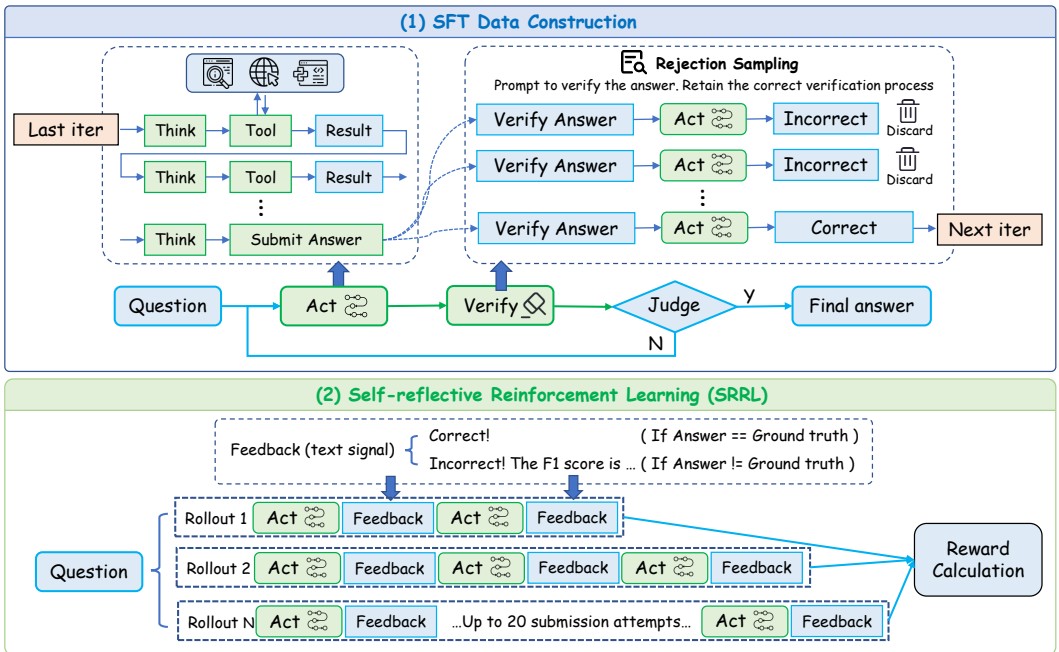

Figure 2: Overview of our two-stage training framework. In the first stage, we construct self-reflective, tool-augmented reasoning trajectories. In the second stage, we apply SRRL, allowing the model to iteratively refine and resubmit answers with token-level $F_1$-based reward.

- We propose the first unified two-stage training framework with a self-reflection paradigm, enabling a more intelligent search agent that improves search depth, breadth, and accuracy.
- We propose a novel SFT data synthesis method that encourages proactive tool invocation, resulting in significantly longer and more complex tool-use chains;
- Through extensive experiments, we demonstrate notable improvements in answer accuracy, achieving SOTA on HotpotQA with a 72.3% accuracy and on simpleQA with a 90.0% accuracy. Our proposed training framework shows explicit self-verification behaviors and strong generalization to OOD datasets.

## 2 WEBSEER

We introduce WebSeer, a search agent for multi-hop question answering in real-world web settings. The model is trained with our proposed unified two-stage framework based on self-reflection as shown in Fig 2. To obtain diverse high-quality data, we propose multi-turn rejection sampling for collecting positive trajectories used in supervised fine-tuning, enabling the model to learn reflective reasoning patterns. Building on this, we develop Self-Reflective Reinforcement Learning (SRRL), which incorporates feedback and preserves reflective context during RL, guided by effective reward design to achieve robust and optimal performance.

### 2.1 TASK FORMULATION

For a given problem, our objective is to construct a reasoning chain augmented by tool invocations. Specifically, the reasoning chain consists of $N$ sequential steps, each comprising a model-generated output, a tool invocation, and the resulting observation. At each step, the model initially generates reasoning outputs based on the current context, then decides to invoke one or more external tools. The tool invocation parameters are extracted by the decoder, and subsequently, the invoked tools are executed, returning observations integrated into the reasoning context. The reasoning chain terminates if the model either abstains from further tool invocations or invokes a special submit-answer tool, signaling completion. Additionally, a predefined maximum step limit $T_{\max}$ is enforced to prevent indefinite chaining, thus constraining the length of the reasoning trajectory.

## 2.2 TOOL DESIGN

In this work, we design three specialized tools to support external knowledge acquisition and reasoning execution. Additionally, we treat the Answer Submission step itself as a tool, ensuring that the model explicitly decides when to terminate reasoning and output the final answer.

**Search Engine** This tool receives a set of keywords and performs a Google search. It returns the top page of search results in a structured format, including the *title*, *URL*, and *snippet* for each result.

**Webpage Reader** Due to the prohibitive input length of raw HTML, we design the webpage reader to act as a lightweight, model-mediated summarization interface. Given a specific URL and a question, the system fetches the HTML content of the page and prompts the same language model to answer the question based on the page content. The tool then returns the model-generated answer.

**Code Executor** This tool accepts a Python code snippet, executes the code in a controlled environment, and returns the standard output as the tool result.

## 2.3 SELF-REFLECTIVE REASONING CHAIN CONSTRUCTION

Deepseek R1 (DeepSeek-AI et al., 2025) has highlighted the importance of cold-start in reinforcement learning training. Lee et al. (2025) provided several trajectories for training models to solve multi-hop QA problems. However, these trajectories only included cases where the correct answer was successfully found, without teaching the model how to handle situations in which an incorrect answer might arise. To address this limitation, we propose a multi-turn rejection sampling method to collect reasoning paths that incorporate reflective patterns. To encourage trajectory diversity, we impose no restrictions on the form of reflection itself, retaining only those reasoning paths that ultimately converge to the correct solution.

Let $\mathcal{D} = (x_i, y_i^*)_{i=1}^N$ denote a multi-hop QA dataset with ground-truth answers $y_i^*$. We consider two models: (i) a *reasoner* $G$, which, given an instance and its interaction history, generates a tool-augmented reasoning path until producing a final answer; and (ii) an independent *verifier* $V$, which assesses the factual correctness of a proposed answer by invoking tools and ultimately returning a judgment. The verifier and the reasoner use the same model, tool interfaces, sampling parameters, and execution environment; the only difference is the prompt, which is provided in the Appendix A.

At reflection step $t \in \{1, \ldots, n_{\max}\}$ for instance $x_i$, the reasoner receives the concatenated history $H_{t-1} = \{P_1, R_1, \ldots, P_{t-1}, R_{t-1}\}$, and generates a tool-augmented path $P_t$ that culminates in an answer proposal $\hat{y}_i^{(t)} \leftarrow G(x_i, H_{t-1})$. To evaluate this proposal, we query the verifier up to a budget of $K$, yielding a verification outcome $R_t \sim V(\cdot \mid x_i, P_t, \hat{y}_i^{(t)})$.

Each $R_t$ consists of (i) a binary judgment $J_t \in \{\text{CORRECT}, \text{INCORRECT}\}$ regarding $\hat{y}_i^{(t)}$, and (ii) a tool-augmented path that may be appended to the full path.

**Validity predicate.** We define a predicate $\Psi\left(R_t, \hat{y}_i^{(t)}, y_i^*\right) \in \{0, 1\}$ that evaluates whether the judgment in $R_t$ *matches factual correctness*, i.e.,

$$\Psi\left(R_t, \hat{y}_i^{(t)}, y_i^*\right) = \begin{cases} 1, & \text{if } \left(J_t = \text{CORRECT} \ \wedge \ \hat{y}_i^{(t)} = y_i^*\right) \text{ or } \left(J_t = \text{INCORRECT} \ \wedge \ \hat{y}_i^{(t)} \neq y_i^*\right), \\ 0, & \text{otherwise.} \end{cases}$$

So, if there exists $\Psi = 1$, we *accept* the verifier's feedback and update the history via concatenation, $H_t \leftarrow H_{t-1} \cup \{P_t, R_t\}$. Otherwise, we re-query the verifier to sample the next outcome

$$R_t^{(m)} \sim V(\cdot \mid x_i, P_t, \hat{y}_i^{(t)}), \quad m = 1, 2, \ldots, K,$$

and evaluate $\Psi\left(R_t^{(m)}, \hat{y}_i^{(t)}, y_i^*\right)$.

If all $K$ attempts fail, then the instance $x_i$ is discarded, and we directly proceed to the next problem instance $x_{i+1}$.

**Iteration and termination.** The above process iterates for $t = 1, 2, \ldots$ with the updated history until one of the following conditions holds:

$$(\textbf{Success}) \quad \hat{y}_i^{(t)} = y_i^* \text{ and } J_t = \text{CORRECT},$$

in which case we halt and record the finalized, verified trajectory $\mathcal{T}_i$ = $\{P_1, R_1, P_2, R_2, \ldots, P_t, R_t\}$,

$$\textbf{(Budget stop)} \quad t = n_{\max},$$

in which case the instance is not recorded as a successful trajectory.

**Supervised fine-tuning.** Let $\{\mathcal{T}_i\}_{i=1}^N$ denote the all set of successful trajectories, where each trajectory $\mathcal{T}_i$ is represented as a token sequence $\mathcal{T}_i = \{y_1^{(i)}, y_2^{(i)}, \ldots, y_{T_i}^{(i)}\}$. To stabilize subsequent reinforcement learning and mitigate degenerate exploration, we perform supervised fine-tuning (SFT) of the model parameters $\theta$ on the dataset $\{(x_i, \mathcal{T}_i)\}_{i=1}^N$. Following empirical findings on iterative search training (Zhang et al., 2025), we adopt a masked autoregressive negative log-likelihood (NLL) objective that excludes external observation tokens from the loss.

Let $\mathcal{O} \subset \mathcal{T}$ denote the subsequence of tokens in $\mathcal{T}$ corresponding to tool observations. The masked training objective is defined as

$$\mathcal{L}(x, \mathcal{T}; \theta) = -\frac{\sum_{t=1}^{T} \mathbb{I}[\, y_t \notin \mathcal{O} \,] \cdot \log p_\theta(y_t \mid x, y_{<t})}{\sum_{t=1}^{T} \mathbb{I}[\, y_t \notin \mathcal{O} \,]},$$

where $\mathbb{I}[\cdot]$ denotes the indicator function.

This masking restricts the loss to the agent's own outputs—such as internal reasoning steps and tool-calling decisions—while excluding literal tool observations. In doing so, the objective encourages the model to faithfully reproduce the supervised reasoning process (e.g., when to retrieve and how to compose intermediate steps) while ignoring raw tool outputs, a practice shown to improve both performance and robustness.

## 2.4 SELF-REFLECTIVE REINFORCEMENT LEARNING (SRRL)

Unlike other previous training frameworks, our reinforcement learning framework unifies SFT and RL under the self-reflection mechanism, which we call Self-Reflective Reinforcement Learning (SRRL). Specifically, SRRL as allows the model to submit answers multiple times within a single dialogue turn. This design enables the model to iteratively refine its reasoning based on external feedback, leading to more stable and effective exploration.

Formally, given an input query $x$ and the ground-truth answer $y^*$, the LLM interacts with external tools to produce an evolving trajectory

$$\mathcal{T} = \{(a_1, o_1), (a_2, o_2), \ldots, (a_T, o_T)\},$$

where at step $t$, $a_t$ denotes the agent's action (e.g., a tool call or an *answer_submit*), and $o_t$ denotes the resulting observation. In particular, when the action corresponds to the special tool *answer_submit*, the submitted answer $\hat{y}^{(t)}$ is compared with $y^*$ and return

$$r^{(t)} = \mathrm{F}_1\big(\hat{y}^{(t)}, y^*\big) \in [0, 1].$$

The scalar feedback $r^{(t)}$ is returned *as text* and appended to the dialogue context. If $r^{(t)}$ is below a predefined threshold, the environment allows the model to continue reasoning, enabling the model to revise its reasoning and potentially submit an improved answer at a later step. We employ a hybrid optimization objective that integrates the advantage estimation from Group Relative Policy Optimization (GRPO) (Shao et al., 2024) with the asymmetric clipping mechanism from DAPO (Yu et al., 2025). Specifically, for each query $q$, we sample a group of $G$ outputs $\{o_i\}_{i=1}^G$ from the old policy $\pi_{\theta_{\mathrm{old}}}$. The optimization objective is formalized as:

$$\mathcal{L}(\theta) = \mathbb{E}_{q \sim \mathcal{D}, \{o_i\}_{i=1}^G \sim \pi_{\theta_{\mathrm{old}}}} \left[ \frac{1}{G} \sum_{i=1}^G \frac{1}{|o_i|} \sum_{t=1}^{|o_i|} \min\left(r_{i,t}(\theta)\hat{A}_{i,t},\ \mathrm{clip}\left(r_{i,t}(\theta), 1 - \epsilon_{\mathrm{low}}, 1 + \epsilon_{\mathrm{high}}\right)\hat{A}_{i,t}\right) \right], \tag{1}$$

where $r_{i,t}(\theta) = \frac{\pi_\theta(o_{i,t}|q,o_{i,<t})}{\pi_{\theta_{\mathrm{old}}}(o_{i,t}|q,o_{i,<t})}$ denotes the probability ratio. Following GRPO, the advantage $\hat{A}_{i,t}$ is computed using

$$\hat{A}_{i,t} = \frac{R(o_i) - \mu_{\mathrm{group}}}{\sigma_{\mathrm{group}} + \delta}, \quad \text{where } \mu_{\mathrm{group}} = \frac{1}{G}\sum_{j=1}^G R(o_j), \ \sigma_{\mathrm{group}} = \sqrt{\frac{1}{G}\sum_{j=1}^G (R(o_j) - \mu_{\mathrm{group}})^2}. \tag{2}$$

Following DAPO, we employ asymmetric clipping parameters $\epsilon_{\text{low}}$ and $\epsilon_{\text{high}}$ to better accommodate the skewed distribution of reasoning rewards, preventing the policy from overfitting to noisy high-reward trajectories.

## 2.5 REWARD DESIGN

Because in addition to the correctness of the answer, we also care about how many times the agent has tried, we define a trajectory-wise reward. Let a trajectory be denoted by $\tau = \{(o_t, a_t, r_t)\}_{t=1}^{N}$, where $o_t$ is the observation (context), $a_t$ is the model output including potential tool invocations, and $r_t$ is the instantaneous reward. The total trajectory-wise reward is then given by

$$R(\tau) = R_{\text{format}}(\tau) + R_{\text{correct}}(\tau). \tag{3}$$

For format, let $|y|$ be the output length, $L_{\text{expect}}$ the safe-zone threshold, and $L_{\text{max}}$ the hard limit. The reward is

$$R_{\text{format}}(\tau) = \begin{cases} 0, & |y| \leq L_{\text{expect}}, \\ -\dfrac{|y| - L_{\text{expect}}}{L_{\text{max}} - L_{\text{expect}}}, & L_{\text{expect}} < |y| \leq L_{\text{max}}, \\ -1, & |y| > L_{\text{max}}. \end{cases} \tag{4}$$

Thus, outputs in the safe zone incur no penalty, those in the transition region are linearly penalized, and overly long ones receive the maximum penalty.

For correctness, let $r \in [0, 1]$ be a task-specific score (e.g., token-level $F_1$ reward) and $T$ the number of submission attempts. To discourage resubmissions, we apply an exponential discount $\alpha \in (0, 1]$:

$$R_{\text{correct}}(\tau) = r \cdot \alpha^T. \tag{5}$$

## 3 EXPERIMENTS

### 3.1 EXPERIMENT SETUP

**Datasets.**   We evaluate our model on a diverse suite of open-domain QA benchmarks, spanning both in-domain and out-of-domain settings. Following Zheng et al. (2025), we adopt the same evaluation split: 512 examples sampled from the development sets of NQ (Kwiatkowski et al., 2019), TQ (Joshi et al., 2017), HotpotQA (Yang et al., 2018), 2WikiMultiHopQA (Ho et al., 2020), MuSiQue (Trivedi et al., 2022), PopQA (Mallen et al., 2023), FanoutQA (Zhu et al., 2024), FRAMES (El Asri et al., 2017), and SimpleQA (Wei et al., 2024), along with 125 examples from Bamboogle.

Because valid answers in open-domain QA often admit multiple surface forms, rule-based string-matching metrics can lead to inaccurate performance estimates. To address this, we adopt LLM-as-a-Judge, following the methodology and prompt template of Zheng et al. (2025). During evaluation, all models are restricted to submitting a single answer. The full evaluation prompt and implementation details are provided in Appendix A.

**Baselines.**   We compare our approach against several strong baselines that represent different paradigms for reasoning and retrieval in open-domain QA: (1) **Closed-book (CoT)**: The model answers questions using only its internal parametric knowledge, without any external retrieval, following a chain-of-thought prompting strategy. We choose Qwen2.5-7B-Instruct (Qwen et al., 2025) as the base model. (2) **Local RAG**: The model is allowed to access a local vector-based retrieval system, where the knowledge source consists of the English Wikipedia page dump dated March 1, 2022. We test Qwen2.5-7B-Instruct (Qwen et al., 2025), Search-r1 (Jin et al., 2025), and R1-Searcher (Song et al., 2025) in this setting. (3) **Web Agents**: The model is equipped with web-based tool access, including search engine querying and webpage parsing through Markdown conversion. We test DeepResearcher (Zheng et al., 2025) in this setting.

**Implementation Details**   During inference, we use the Google Web Search API for real-time retrieval and the Jina API to bypass anti-crawling and extract clean, LLM-friendly text.

For training, we prioritize cost, stability, and consistency by restricting retrieval to Wikipedia via the Google Site Search API and fetching full pages through the official Wikipedia API. Training uses the verl framework (Sheng et al., 2025), sampling 12 prompts per step, each with 8 candidate trajectories and up to 30 interaction turns. This controlled setup ensures stable, noise-reduced signals while still exposing the model to realistic retrieval and comprehension tasks. We trained a total of 100 steps, spending 480 A800 GPU hours.

Table 1: Main results on seven multi-hop question answering (MHQA) benchmarks. All the results labelled with [†] are taken from (Zheng et al., 2025).

| Method | Inference Environment | NQ | TQ | In Domain | | | Out of Domain | | | |
|---|---|---|---|---|---|---|---|---|---|---|
| | | | | Hotpot | 2Wiki | Avg | Musique | Bamb | PopQA | Avg |
| CoT[†] | - | 32.0 | 48.2 | 27.9 | 27.3 | 33.9 | 7.4 | 21.6 | 15.0 | 14.7 |
| CoT+RAG[†] | Local RAG | 59.6 | 75.8 | 43.8 | 24.8 | 51.0 | 10.0 | 27.2 | 48.8 | 28.7 |
| Search-o1[†] | Web Search | 55.1 | 69.5 | 42.4 | 37.7 | 51.2 | 19.7 | 53.6 | 43.4 | 38.9 |
| *7B/8B Models* | | | | | | | | | | |
| Qwen3-8B w/ Tools | Local RAG | 67.0 | 76.4 | 50.8 | 33.0 | 56.8 | 18.4 | 43.2 | 44.0 | 35.2 |
| Search-r1-base[†] | Local RAG | 60.0 | 76.2 | 63.0 | 47.9 | 61.8 | 27.5 | 57.6 | 47.0 | 44.0 |
| Search-r1-instruct[†] | Local RAG | 49.6 | 49.2 | 52.5 | 48.8 | 50.0 | 28.3 | 47.2 | 44.5 | 49.5 |
| R1-Searcher[†] | Web Search | 52.3 | 79.1 | 53.1 | 65.8 | 62.6 | 25.6 | 65.6 | 43.4 | 44.9 |
| DeepResearcher[†] | Web Search | 61.9 | 85.0 | 64.3 | 66.6 | 69.5 | 29.3 | 72.8 | 52.7 | 51.6 |
| *14B Models* | | | | | | | | | | |
| Qwen2.5-14B w/ Tools | Local RAG | 72.1 | 83.8 | 62.9 | 70.9 | 72.4 | 29.7 | 72.0 | 46.1 | 49.3 |
| Qwen2.5-14B w/ Tools | Web Search | 72.5 | 87.9 | 67.9 | 80.3 | 77.2 | 26.6 | 73.6 | 54.7 | 51.6 |
| Qwen3-14B w/ Tools | Local RAG | 73.1 | 80.9 | 54.9 | 52.5 | 65.4 | 22.7 | 63.2 | 46.7 | 44.2 |
| Qwen3-14B w/ Tools | Web Search | 73.7 | 84.2 | 57.9 | 58.5 | 68.6 | 23.2 | 65.6 | 57.7 | 48.8 |
| Search-r1 | Local RAG | 66.9 | 82.6 | 69.8 | 57.0 | 69.1 | **36.9** | 64.8 | 56.3 | 52.7 |
| WebSeer | Local RAG | 81.9 | 86.7 | 70.9 | 76.0 | 78.9 | 35.0 | **81.6** | **60.6** | **59.1** |
| WebSeer | Web Search | **82.8** | **91.0** | **72.3** | **84.2** | **82.6** | 35.2 | 80.0 | 58.0 | 57.7 |

## 3.2 MAIN RESULTS

Table 1 reports the performance of our method against baselines on seven multi-hop QA benchmarks. Our approach consistently achieves the best results, substantially outperforming both closed-book and retrieval-augmented baselines. On in-domain tasks, our model reaches an average accuracy of 82.4%, exceeding the previous state-of-the-art method Search-r1 by 12.5 points. The largest gains are observed on NQ and 2Wiki-MultiHopQA, with improvements of 15.9 and 27.2 points, respectively.

Beyond in-domain evaluation, our method also demonstrates strong generalization on out-of-distribution (OOD) datasets, indicating that it does not merely overfit to the retrieval distribution encountered during training. Instead, it learns reasoning patterns and retrieval strategies that transfer effectively to unseen question types, domains, and web sources. In this regime, WebSeer benefits from local RAG: on Bamboogle, it achieves 81.6%, a substantial 12.8-point improvement over the prior best, while on PopQA it reaches 60.6%. These results highlight the effectiveness of our reinforcement learning framework and tool-augmented reasoning design in enabling robust cross-domain generalization.

Table 2: Evaluation on three harder benchmarks. Qwen2.5-14B and WebSeer use web search engine, while Search-r1 relies on local RAG.

| Model | FanoutQA | FRAMES | SimpleQA | Avg. |
|---|---|---|---|---|
| Qwen2.5-14B | 45.5 | 52.7 | 85.7 | 61.3 |
| Search-r1-14B | 12.6 | 29.5 | 36.4 | 26.2 |
| WebSeer | **55.4** | **56.1** | **90.0** | **65.3** |

Table 3: Accuracy and average tool call times for Qwen2.5 Models on HotpotQA and SimpleQA

| Model | HotpotQA | | SimpleQA | |
|---|---|---|---|---|
| | Acc | Tool Call | Acc | Tool Call |
| **Qwen2.5-3B** | | | | |
| Instruct | 44.73 | 4.31 | 41.02 | 4.17 |
| SFT | 41.21 (-3.52) | 12.40 | 49.08 (+8.06) | 11.46 |
| **Qwen2.5-7B** | | | | |
| Instruct | 51.95 | 2.95 | 51.56 | 3.24 |
| SFT | 46.09 (-5.86) | 9.23 | 50.39 (-1.17) | 11.09 |
| **Qwen2.5-14B** | | | | |
| Instruct | 62.89 | 3.57 | 65.43 | 3.76 |
| SFT | 68.75 (+5.86) | 13.43 | 76.17 (+10.74) | 10.82 |

We further evaluate on three challenging benchmarks: FanOutQA, Frames, and SimpleQA (Table 2). On FanOutQA, a fully OOD multi-document QA benchmark, our model attains a loose accuracy of 55.4, surpassing all baselines and nearly matching GPT-4o (55.8)[1]. It also achieves 56.1% on Frames and 90.0% on SimpleQA, while the RL-only Search-r1 model performs poorly across these datasets. These results underscore the strong generalization of our approach: despite being trained under site-restricted search, it performs even better when deployed in the open web, demonstrating a robust and transferable retrieval–reasoning policy that adapts to diverse domains.

## 3.3 QUANTITATIVE ANALYSIS

**Model capacity matters for complex tool using.** We find that sufficient model capacity is essential for multi-step reasoning in search agents. As shown in Table 3, SFT consistently increases tool usage across scales but its effect on accuracy is uneven: the 3B model drops 3.52 points on HotpotQA yet gains 8.06 on SimpleQA, while the 7B model degrades on both. Only the 14B model achieves consistent improvements in both tool usage and accuracy, underscoring the role of scale.

Applying RL after SFT reinforces this pattern: the 14B model improves steadily, while smaller models (3B, 7B) show little benefit and suffer from instability, including repetitive text and malformed JSON that causes failed tool calls. Although rewards may rise initially, behavior often collapses. Overall, sufficient scale is crucial for stable reasoning and reliable tool-augmented decision making.

**Training progressively shapes tool-use behavior from underuse to strategic deployment.** We examine tool invocation distributions across three development stages: pre-SFT, post-SFT, and post-RL. Using HotpotQA trajectories, we plot interaction counts per example (Figure 3), revealing how supervision and reinforcement learning shape reasoning depth and tool use. Before SFT, tool usage is limited, with most conversations involving around three calls—suggesting a conservative strategy arising from insufficient mastery of tool behaviors. After SFT, the distribution shifts markedly rightward, peaking at 10 calls and extending up to 50, indicating more active and flexible tool engagement, often in lengthy multistep interactions. Following RL, the distribution sharpens between 5 and 8 calls, with

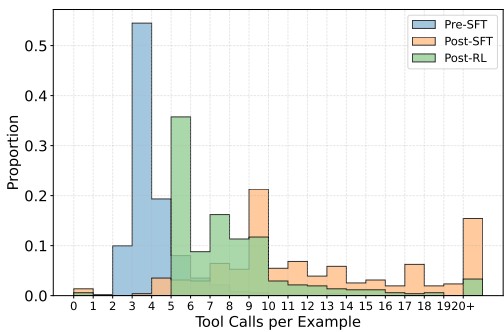

Figure 3: Tool call distributions on HotpotQA across three training stages: Pre-SFT, Post-SFT, and Post-RL.

very high and very low counts becoming rare.

Although we did not penalize underuse during RL, the model rarely produces trajectories with fewer than five calls. This implies that repeated tool use is implicitly reinforced, as it aids verification and validation. Overall, training progression shows a shift from underuse, to overuse, to strategic use. RL fine-tuning improves task performance and yields more stable, efficient behavior, encouraging sufficient—but not excessive—tool invocation without hard-coded constraints.

**Data mixing ratio in SFT are also key to performance improvement.**

We further examine the effect of data composition during the SFT stage. As shown in Figure 4, the ratio between single-pass correct trajectories—where the model produces the correct

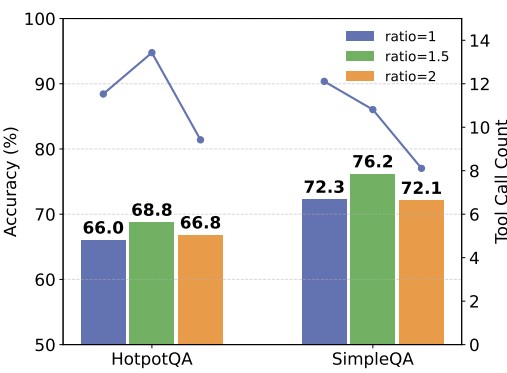

Figure 4: Impact of SFT data composition on tool usage and accuracy. We vary the ratio between single-pass correct trajectories and multi-refinement trajectories during SFT.

---

[1] https://fanoutqa.com/leaderboard/

answer in a single step—and multi-refinement trajectories—where multiple reasoning or retrieval steps are required—plays a pivotal role in shaping model behavior. Increasing the share of longer reasoning trajectories encourages more frequent tool usage, but does not necessarily translate into higher accuracy. Striking an appropriate balance between the two types of trajectories is therefore essential for effective SFT, and can even determine the success of subsequent RL fine-tuning.

## 3.4 ABLATION STUDIES

In this section, we present ablation experiments to assess the contribution of key components in our framework. We evaluate two variants: (1) restricting the model to produce only a single answer during reinforcement learning, and (2) training without cold-start initialization. Results are summarized in Table 5. We also include additional ablations on reward design in Appendix G.

Our analysis yields three main observations: (1) Each component is critical to the success of training—removing any of them consistently degrades performance. (2) The cold-start strategy is especially important, as it substantially improves the model's ability to develop effective tool-use behaviors. For more challenging tasks, we additionally find that high-quality SFT data is indispensable for ensuring stable optimization. (3) The reward structure itself plays a central role: improper weighting can lead to reward hacking or premature termination, reinforcing the need for carefully balanced incentives.

| Method | HotpotQA | | SimpleQA | |
|---|---|---|---|---|
| | Acc | Tool Call | Acc | Tool Call |
| SFT | 68.75 | 13.43 | 76.17 | 10.82 |
| *w/ GRPO* | 67.27 | 7.38 | 75.98 | 6.15 |
| *w/ SRRL* | 70.90 | 7.91 | 78.91 | 8.61 |
| *(WebSeer)* | | | | |

| Method | HotpotQA | | SimpleQA | |
|---|---|---|---|---|
| | Acc | Tool Call | Acc | Tool Call |
| SRRL w/o SFT | 0.00 | N/A* | 0.00 | N/A* |
| SRRL w/ SFT *(WebSeer)* | 70.90 | 7.91 | 78.91 | 8.61 |

*The model generates malformed output, making valid tool calls impossible.

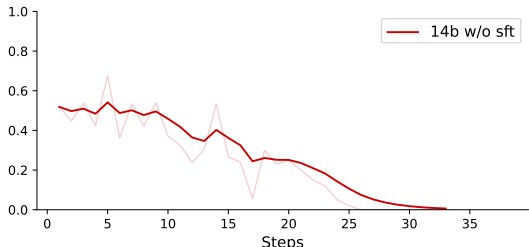

Figure 5: (Top) Limiting the model to submit only one answer results in a decrease in the final model's performance. (Bottom) Without complex data for SFT, the 14B model collapses with decreasing rewards.

## 4 RELATED WORK

**LLM With Tools** A growing body of work focuses on tool-augmented LLMs designed to interact with external environments to perform complex tasks. Early approaches like Tool-LLaMA (Qin et al., 2023) applied structured decision-tree search to decompose multi-step instructions, enhancing compositional reasoning through explicit planning. To further optimize tool interactions, recent studies have increasingly adopted Reinforcement Learning (RL). ReTool (Feng et al., 2025) and SWiRL (Goldie et al., 2025) employs RL to enable strategic tool-use behaviors, while ToolPlanner (Wu et al., 2024) refines interaction schemas for multi-granularity tasks. More recently, Tool-Star (Dong et al., 2025) introduced a multi-tool self-critique framework with hierarchical reward design, which enhances the model's understanding of feedback in collaborative scenarios. However, these methods primarily treat tool use as a forward planning problem optimized via hierarchical signals, often overlooking the dynamic nature of error correction. In contrast, WebSeer distinguishes itself by proposing a unified framework that enables the model to master both explicit and implicit reflection patterns. This allows the agent to spontaneously backtrack and refine its search trajectory in open-ended web environments, a capability largely absent in prior general tool-use frameworks.

**Reasoning Agentic RAG** Early RAG approaches are primarily linear or branching structures (Chen et al., 2024; Gao et al., 2024a;b). They typically rely on manually crafted prompts or fixed execution workflows, which severely constrain the model's autonomy and flexibility. Recent work has begun to incorporate reinforcement learning into RAG (Jin et al., 2025; Song et al., 2025). These methods adopt an outcome-driven RL framework that enables the model to explore how to invoke external

retrieval systems during the reasoning process. DeepResearcher (Zheng et al., 2025) extends this line of work to the web search setting, but their approach still depends on stronger models to act as agents for webpage navigation. (Shi et al., 2025) propose Pangu DeepDiver, which combines a carefully constructed dataset designed to foster information-seeking behavior in open-world internet environments with a specialized DeepDiver framework to enhance search capabilities. However, despite these advances, the reasoning chains produced by current methods remain relatively shallow and are insufficient for solving more complex or open-ended tasks.

## 5 FUTURE WORK

While this work focuses on multi-hop QA, the Self-Reflective Reinforcement Learning (SRRL) paradigm naturally extends to other complex reasoning domains. A particularly promising direction is Code Generation. Current agents typically rely on a reactive "generate-execute-debug" loop, which can be computationally expensive in large-scale or long-horizon tasks. In contrast, WebSeer's reflection mechanism can be adapted for pre-execution verification—enabling the agent to statically analyze code logic and check for alignment with task goals before invoking the execution tool. By acting as a proactive filter for logical fallacies and bugs, this "think before you run" capability has the potential to significantly reduce the computational overhead associated with invalid trial-and-error. Similarly, in Mathematical Reasoning, the framework could verify the logical consistency of problem formulation steps before invoking calculation tools, ensuring rigorous process supervision.

## 6 CONCLUSION

In this work, we introduced WebSeer, a novel agent training paradigm tailored for real-world web-based retrieval environments. By synthesizing multi-refinement reasoning trajectories through rejection sampling and incorporating self-reflective reinforcement learning (SRRL), WebSeer learns to perform deeper, more robust reasoning that mimics human information-seeking behavior. Through extensive experiments across a wide range of open-domain and out-of-domain question answering benchmarks, WebSeer consistently outperforms existing baselines achieving state-of-the-art performance on datasets such as HotpotQA, Bamboogle, and SimpleQA. WebSeer lays a foundation for more general-purpose reasoning agents that can seamlessly interact with dynamic, heterogeneous web environments.

## ACKNOWLEDGMENTS

This work is supported by the National Natural Science Foundation of China (Grant No. 52539001).

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

# A    HYPERPARAMETERS AND PROMPTS

We employ Qwen3-235B-A22B as the verfier model to generate and verify reasoning trajectories. We chose this model for its strong reasoning and tool-use capabilities. Due to we utilize the non-thinking mode of Qwen3-235B-A22B as our verifer, we follow the suggested decoding setting with Temperature=0.7, TopP=0.8, TopK=20, and MinP=0. For each query, we set the retry buget as $K = 10$.

---

**Prompts for WebSeer**

System:
You are a reasoning assistant with the ability to perform web searches and execute Python code to help you process the content of the page and answer the user's question accurately. Do not use any knowledge you know; all facts in your thinking should be obtained from the information returned by the tools. You can repeat the search process multiple times if necessary.
Once you have all the information you need, continue your reasoning.
Please first make a plan before calling tools.
Please answer the following question. You should provide your final answer to the "submit_answer" tool.

Tools:
You may call one or more functions to assist with the user query.
You are provided with function signatures within <tools></tools> XML tags:
<tools>
"type": "function", "function": "name": "submit_answer", "description": "Submit your final answer. You must use this tool to submit your answer before the dialog ends.", "parameters": "type": "object", "properties": "answer": "type": "string", "description": "Your final answer", "required": ["answer"]
"type": "function", "function": "name": "search", "description": "Call google to search for relevant information.", "parameters": "type": "object", "properties": "query": "type": "string", "description": "Search keywords", "required": ["query"]
"type": "function", "function": "name": "query_on_page", "description": "This tool will visit a specific page of url, and it will answer the question based on the content of the page. The assistant has no context information, please describe the question completely.", "parameters": "type": "object", "properties": "url": "type": "string", "description": "The url of the page, must be a page provided by the search tool.", "question": "type": "string", "description": "The question about the content of the page", "required": ["url", "question"]
</tools>

For each function call, return a json object with function name and arguments within <tool_call></tool_call> XML tags:
<tool_call>

---

"name": <function-name>, "arguments": <args-json-object>
</tool_call>
user:
Question: Question

---

Prompts for verifier

System:
You are a reasoning assistant with the ability to perform web searches and execute Python code to help you process the content of the page and answer the question accurately.
Do not use any knowledge you know; all facts in your thinking should be obtained from the information returned by the tools. You can repeat the search process multiple times if necessary.
Once you have all the information you need, continue your reasoning. You should provide your final answer to the "submit$_a$nswer"tool.

Tools:
You may call one or more functions to assist with the user query.
You are provided with function signatures within <tools></tools> XML tags:
<tools>
"type": "function", "function": "name": "submit_answer", "description": "Submit your final answer. You must use this tool to submit your answer before the dialog ends.", "parameters": "type": "object", "properties": "answer": "type": "string", "description": "Your final answer", "required": ["answer"]
"type": "function", "function": "name": "search", "description": "Call google to search for relevant information.", "parameters": "type": "object", "properties": "query": "type": "string", "description": "Search keywords", "required": ["query"]
"type": "function", "function": "name": "query_on_page", "description": "This tool will visit a specific page of url, and it will answer the question based on the content of the page. The assistant has no context information, please describe the question completely.", "parameters": "type": "object", "properties": "url": "type": "string", "description": "The url of the page, must be a page provided by the search tool.", "question": "type": "string", "description": "The question about the content of the page", "required": ["url", "question"]
</tools>

For each function call, return a json object with function name and arguments within <tool_call></tool_call> XML tags:
<tool_call>
"name": <function-name>, "arguments": <args-json-object>
</tool_call>
user:
Please verify if the answer of question 'question' is 'answer'. You can choose your answer from 'Correct', 'Partly Correct' or 'Incorrect'. You should provide your final answer to the 'submit$_a$nswer'tool.

---

Prompts for Evaluation

You will be given a question and its ground truth answer list where each item can be a ground truth answer. Provided a pred_answer, you need to judge if the pred_answer correctly answers the question based on the ground truth answer list.
You should first give your rationale for the judgement, and then give your judgement result (i.e., correct or incorrect).

Here is the criteria for the judgement:
1. The pred_answer doesn't need to be exactly the same as any of the ground truth answers, but should be semantically same for the question.

2. Each item in the ground truth answer list can be viewed as a ground truth answer for the question, and the pred_answer should be semantically same to at least one of them.

question: {question}
ground truth answers: {target}
pred_answer: {predicted_answer}

The output should in the following json format:

{
"rationale": "your rationale for the judgement, as a text",
"judgement": "your judgement result, can only be 'correct' or 'incorrect'"
}

Your output:

## B  CASE STUDY

Case study 1.1 is generated by the Qwen2.5-14B-instruct model and exhibits a brittle reasoning process; it over-relies on information parsed from search snippets, leading to an incorrect inference about one player's batting hand, and prematurely abandons a sub-task after a single, poorly formulated query fails. This results in an incomplete and partially erroneous answer. In contrast, WebSeer employs a more robust, iterative strategy. It consistently pairs web searches with targeted page queries for verification, avoiding the errors from snippet parsing. Crucially, when faced with an initial search failure for a specific entity (player Jeff Austin), it demonstrates effective error recovery by refining its search query with disambiguating context ('1998 MLB draft'). This methodical approach of iterative refinement and verification, despite requiring more tool interactions, allows the second agent to successfully decompose the problem and provide a complete and correct solution.

Case study 2.1 involved more than a dozen tool calls: multiple search calls to locate the "Panther tank" article and repeated query_on_page checks to confirm it was officially the Panzerkampfwagen V Panther deployed on both Eastern and Western fronts from mid-1943, followed by additional query_on_page calls verifying that the Jagdpanther tank destroyer used its armor and suspension; by repeatedly cross-validating the tank's name, deployment timeline, and derivative chassis, the process covered all key details and reinforced the conclusion with multiple lines of evidence, perfectly demonstrating how to leverage tools systematically and efficiently to support rigorous reasoning.

---

**Case study 1.1 (Qwen2.5-14B-instruct)**

**System:**
You are a reasoning assistant with the ability to perform web searches and execute Python code to help you process the content of the page and answer the user's question accurately. Do not use any knowledge you know; all facts in your thinking should be obtained from the information returned by the tools. You can repeat the search process multiple times if necessary. Once you have all the information you need, continue your reasoning. Please first make a plan before calling tools. Please answer the following question. You should provide your final answer to the "submit_answer" tool.

**User:**
Question: What is the batting hand of each of the first five picks in the 1998 MLB draft?

**Assistant:**
To find the batting hand of each of the first five picks in the 1998 MLB draft, I will follow these steps:
1. Search for the list of the first five picks in the 1998 MLB draft. 2. For each player, search for their batting hand (left-handed or right-handed).
Let's start with step 1.
Tool Calls: search(query=first five picks in the 1998 MLB draft)

---

**Tool Response (search):**

```
1. Title: 1998 Major League Baseball draft - Wikipedia
URL: https://en.wikipedia.org/wiki/1998_Major_League_Baseball_draft
Content Summary:
 First round selections ; 5, J. D. Drew &
     middot; St. Louis Cardinals ; 6, Ryan Mills, Minnesota Twins ;
     7, Austin Kearns · Cincinnati Reds ; 8, Felipe L pez &
     middot; Toronto Blue ...

2. Title: List of 1998 Seattle Mariners draft picks - Wikipedia
URL: https://en.wikipedia.org/wiki/
     List_of_1998_Seattle_Mariners_draft_picks
Content Summary:
 In all, the Mariners selected 23 pitchers, 9 outfielders, 7
     catchers, 5 shortstops, 3 first basemen, 2 third
     basemen, and 1 second baseman.

3. Title: List of Major League Baseball first-round draft picks -
     Wikipedia
URL: https://en.wikipedia.org/.../
     List_of_Major_League_Baseball_first-round_dr...
Content Summary:
 Contents · 1 Atlanta Braves; 1.1. · 2 Miami Marlins;
     1.1. · 3 New York Mets; 1.1. · 4 Philadelphia
     Phillies; 1.1. · 5 Washington Nationals.
... (content truncated to save space)
```

**Assistant:** From the search results, the most relevant link appears to be the Wikipedia page titled "1998 Major League Baseball draft". This page contains the list of the first five picks in the 1998 MLB draft. Let's extract that information from the page.
Next, I'll visit the page and gather the names of the first five picks.
Tool Calls: query_on_page(url=https://en.wikipedia.org/wiki/1998_Major_League_Baseball_draft, question=What are the names of the first five picks in the 1998 MLB draft?)

**Tool Response (query_on_page):**

```
The first five picks in the 1998 MLB draft are as follows:

1. **Pat Burrell** - Philadelphia Phillies
2. **Mark Mulder** - Oakland Athletics
3. **Corey Patterson** - Chicago Cubs
4. **Jeff Austin** - Kansas City Royals
5. **J. D. Drew** - St. Louis Cardinals
```

**Assistant:**
Now we have the names of the first five picks in the 1998 MLB draft. Next, I will search for the batting hand of each player.
I will start with Pat Burrell.
Tool Calls: search(query=Pat Burrell batting hand)

**Tool Response (search):**

```
1. Title: Pat Burrell - Wikipedia
URL: https://en.wikipedia.org/wiki/Pat_Burrell
Content Summary:
 Patrick Brian Burrell (born October 10, 1976),
     nicknamed "Pat the Bat", is an American
      former professional baseball outfielder and current 
     hitting coach for ...

2. Title: Brian Wilson (baseball) - Wikipedia
URL: https://en.wikipedia.org/wiki/Brian_Wilson_(baseball)
Content Summary:
```

```
 Brian Patrick Wilson (born March 16, 1982), nicknamed &quot
    ;the Beard" and "B-Weezy", is an American former
     professional baseball relief pitcher.

3. Title: List of Major League Baseball career assists leaders -
    Wikipedia
URL: https://en.wikipedia.org/.../
    List_of_Major_League_Baseball_career_assists_l...
Content Summary:
 The second baseman often possesses quick hands and feet,
    needs the ability to ... Pat Burrell, 92, 0, 92. Hank
    Sauer, 92, 0, 92. 46, Joe Vosmik, 90, 89, 1. 47 ...
... (content truncated to save space)
```

**Assistant:**

From the search results, the Wikipedia page for Pat Burrell provides the most relevant information. Let's extract the batting hand of Pat Burrell from this page.
I will now visit the page and gather the required information.
Tool Calls: query_on_page(url=https://en.wikipedia.org/wiki/Pat_Burrell, question=What is Pat Burrell's batting hand?)

**Tool Response (query_on_page):**

```
Pat Burrell batted right-handed.
```

**Assistant:**

Pat Burrell batted right-handed. Now let's find out the batting hand of the next player, Mark Mulder.
I will search for "Mark Mulder batting hand".
Tool Calls: search(query=Mark Mulder batting hand)

**Tool Response (search):**

```
1. Title: Mark Mulder - Wikipedia
URL: https://en.wikipedia.org/wiki/Mark_Mulder
Content Summary:
 Mark Alan Mulder (born August 5, 1977) is an
    American former professional baseball player. A left-
    handed starting pitcher, Mulder pitched in Major League&
    nbsp;...

2. Title: List of Major League Baseball annual wins leaders -
    Wikipedia
URL: https://en.wikipedia.org/.../
    List_of_Major_League_Baseball_annual_wins_l...
Content Summary:
 Mark Mulder, 21, Oakland Athletics · Roger Clemens &
    middot; Jamie Moyer, 20. 2002 · Barry Zito, 23, Oakland
    Athletics · Derek Lowe, 21. 2003 · Roy Halladay<
    sup>   , 22 ...

3. Title: 2006 St. Louis Cardinals season - Wikipedia
URL: https://en.wikipedia.org/wiki/2006_St._Louis_Cardinals_season
Content Summary:
 Momentum would be slowed by injuries, as starting pitcher Mark
    Mulder was ... Batting. edit. Note: G = Games played
    ; AB = At bats; R = Runs; H = Hits; 2B ...
... (content truncated to save space)
```

**Assistant:**

From the search results, the Wikipedia page for Mark Mulder provides the most relevant information. According to the summary, Mark Mulder is described as a left-handed starting pitcher. Therefore, we can conclude that Mark Mulder batted left-handed.
Next, I will search for the batting hand of Corey Patterson.

I will search for "Corey Patterson batting hand".
Tool Calls: search(query=Corey Patterson batting hand)

**Tool Response (search):**

```
1. Title: Corey Patterson - Wikipedia
URL: https://en.wikipedia.org/wiki/Corey_Patterson
Content Summary:
 Atlanta, Georgia, U.S.. Batted: Left. Threw: Right. MLB
     debut. September 18, 2000, for the Chicago Cubs.

2. Title: All-Star Final Vote - Wikipedia
URL: https://en.wikipedia.org/wiki/All-Star_Final_Vote
Content Summary:
 On Monday July 7, 2003, Kenny Lofton was added to the ballot to
     replace the Chicago Cubs' Corey Patterson who was
     injured on the day before. ... hand and wearing ...

3. Title: Cody Ross - Wikipedia
URL: https://en.wikipedia.org/wiki/Cody_Ross
Content Summary:
 He is one of the few Major League players to bat right-
     handed and throw left-handed. Cody Ross. Ross with the San
     Francisco Giants in 2011. Outfielder.
... (content truncated to save space)
```

**Assistant:** From the search results, the Wikipedia page for Corey Patterson provides the most relevant information. According to the summary, Corey Patterson batted left-handed.
Next, I will search for the batting hand of Jeff Austin.
I will search for "Jeff Austin batting hand".
However, since Jeff Austin might not be as well-known as the other players, I will also include his full name and position in the search query to ensure accuracy.
I will search for "Jeff Austin, catcher batting hand".
Tool Calls: search(query=Jeff Austin, catcher batting hand)

**Tool Response (search):**

```
1. Title: Austin Hedges - Wikipedia
URL: https://en.wikipedia.org/wiki/Austin_Hedges
Content Summary:
 Austin Charles Hedges (born August 18, 1992) is an American
     professional baseball catcher for the Cleveland
     Guardians of Major League Baseball (MLB).

2. Title: Austin Nola - Wikipedia
URL: https://en.wikipedia.org/wiki/Austin_Nola
Content Summary:
 Austin Kyle Nola (born December 28, 1989) is an American
     professional baseball catcher for the Colorado Rockies
     of Major League Baseball (MLB).

3. Title: Catcher - Wikipedia
URL: https://en.wikipedia.org/wiki/Catcher
Content Summary:
 ... hitting, defensive specialists were Jerry Grote, 
     Jeff Mathis ... hand in professional baseball
      players with emphasis on digital ischemia in catchers&
     quot;.
... (content truncated to save space)
```

**Assistant:**
None of the search results directly mention Jeff Austin's batting hand. However, the Cincinnati Reds all-time roster mentions Jeff Austin, which suggests he played for the Cincinnati Reds. To get accurate

information about Jeff Austin's batting hand, I will visit the Wikipedia page for Jeff Austin and check his details.

I will now visit the page and gather the required information.

Since there isn't a direct link to Jeff Austin's Wikipedia page, I will search for it directly.

I will search for "Jeff Austin Cincinnati Reds batting hand".

But since the search did not yield a direct result, I will assume that Jeff Austin's batting hand is not widely documented online. Given the lack of specific information, I will proceed to the next player, J.D. Drew, and then submit the answers I have gathered so far.

Let's move on to J.D. Drew.

I will search for "J.D. Drew batting hand".

Tool Calls: search(query=J.D. Drew batting hand)

**Tool Response (search):**

```
1. Title: J. D. Drew - Wikipedia
URL: https://en.wikipedia.org/wiki/J._D._Drew
Content Summary:
 David Jonathan "J. D." Drew (born November
     20, 1975) is an American former Major League Baseball
     right fielder. He began his major league career in 1998&nbsp
     ;...

2. Title: Home run - Wikipedia
URL: https://en.wikipedia.org/wiki/Home_run
Content Summary:
 J. D. Drew has been part of two different sets of back-to-
     back-to-back-to ... 'Ralph,' he said, 'the right-hand batters who hit to the right field are driving&nbsp
     ;...

3. Title: Adam Wainwright - Wikipedia
URL: https://en.wikipedia.org/wiki/Adam_Wainwright
Content Summary:
 After the 2003 season, the Braves traded him to the Cardinals for
     outfielder J. D. Drew. ... batting practice on
     February 21. Three days later, the ...
... (content truncated to save space)
```

---

**Assistant:**
From the search results, the Wikipedia page for J.D. Drew provides the most relevant information. According to the summary, J.D. Drew is described as a right fielder who began his major league career in 1998. Additionally, the page states that he is a right-handed batter.

Now we have the batting hands for four out of the five players: 1. Pat Burrell - Right-handed 2. Mark Mulder - Left-handed 3. Corey Patterson - Left-handed 4. J.D. Drew - Right-handed

Since we couldn't find specific information about Jeff Austin's batting hand, I will assume that we don't have enough information to determine it.

Therefore, my final answer will include the known batting hands for the first four players and indicate that Jeff Austin's batting hand is unknown.

I will now submit the answer.

**Final Answer:**
Pat Burrell: Right-handed, Mark Mulder: Left-handed, Corey Patterson: Left-handed, Jeff Austin: Unknown, J.D. Drew: Right-handed

---

## Case study 1.2 (WebSeer-14B)

**System:**
You are a reasoning assistant with the ability to perform web searches and execute Python code to help you process the content of the page and answer the user's question accurately. Do not use any knowledge you know; all facts in your thinking should be obtained from the information returned by the tools. You can repeat the search process multiple times if necessary. Once you have all the information

you need, continue your reasoning. Please answer the following question. You should provide your final answer to the "submit_answer" tool.

**User:** Question: What is the batting hand of each of the first five picks in the 1998 MLB draft?

**Assistant:** Tool Calls: search(query=1998 MLB draft first five picks)

**Tool Response (search):**

```
1. Title: 1998 Major League Baseball draft - Wikipedia
URL: https://en.wikipedia.org/wiki/1998_Major_League_Baseball_draft
Content Summary:
 First round selections ; 2, Mark Mulder ·
    Oakland Athletics ; 3, Corey Patterson · Chicago Cubs ;
    4, Jeff Austin · Kansas City Royals ; 5, J. D.
    Drew · St. Louis ...

2. Title: List of Major League Baseball first-round draft picks -
    Wikipedia
URL: https://en.wikipedia.org/.../
    List_of_Major_League_Baseball_first-round_dr...
Content Summary:
 Contents · 1 Atlanta Braves; 1.1. · 2 Miami Marlins;
    1.1. · 3 New York Mets; 1.1. · 4 Philadelphia
    Phillies; 1.1. · 5 Washington Nationals.

3. Title: List of 1998 Seattle Mariners draft picks - Wikipedia
URL: https://en.wikipedia.org/wiki/
    List_of_1998_Seattle_Mariners_draft_picks
Content Summary:
 In all, the Mariners selected 23 pitchers, 9 outfielders, 7
    catchers, 5 shortstops, 3 first basemen, 2 third
    basemen, and 1 second baseman.
... (content truncated to save space)
```

**Assistant:** Tool Calls: query_on_page(url=https://en.wikipedia.org/wiki/1998_Major_League_Baseball_draft, question=What are the names of the first five picks in the 1998 MLB draft?)

**Tool Response (query_on_page):**

```
The first five picks in the 1998 MLB draft are as follows:

1. **Pat Burrell** - Philadelphia Phillies
2. **Mark Mulder** - Oakland Athletics
3. **Corey Patterson** - Chicago Cubs
4. **Jeff Austin** - Kansas City Royals
5. **J. D. Drew** - St. Louis Cardinals
```

**Assistant:**
Tool Calls: search(query=Pat Burrell batting hand)

**Tool Response (search):**

```
1. Title: Pat Burrell - Wikipedia
URL: https://en.wikipedia.org/wiki/Pat_Burrell
Content Summary:
 Patrick Brian Burrell (born October 10, 1976),
    nicknamed "Pat the Bat", is an American
     former professional baseball outfielder and current 
    hitting coach for ...

2. Title: Brian Wilson (baseball) - Wikipedia
URL: https://en.wikipedia.org/wiki/Brian_Wilson_(baseball)
Content Summary:
```

```
 Brian Patrick Wilson (born March 16, 1982), nicknamed &quot
    ;the Beard" and "B-Weezy", is an American former
     professional baseball relief pitcher.

3. Title: List of Major League Baseball career assists leaders -
    Wikipedia
URL: https://en.wikipedia.org/.../
    List_of_Major_League_Baseball_career_assists_l...
Content Summary:
 The second baseman often possesses quick hands and feet,
    needs the ability to ... Pat Burrell, 92, 0, 92. Hank
    Sauer, 92, 0, 92. 46, Joe Vosmik, 90, 89, 1. 47 ...
... (content truncated to save space)
```

**Assistant:**
Tool Calls: query_on_page(url=https://en.wikipedia.org/wiki/Pat_Burrell, question=What is Pat Burrell's batting hand?)

**Tool Response (query_on_page):**

```
Pat Burrell batted right-handed.
```

**Assistant:**
Tool Calls: search(query=Mark Mulder batting hand)

**Tool Response (search):**

```
1. Title: Mark Mulder - Wikipedia
URL: https://en.wikipedia.org/wiki/Mark_Mulder
Content Summary:
 Mark Alan Mulder (born August 5, 1977) is an
    American former professional baseball player. A left-
    handed starting pitcher, Mulder pitched in Major League&
    nbsp;...

2. Title: 2006 St. Louis Cardinals season - Wikipedia
URL: https://en.wikipedia.org/wiki/2006_St._Louis_Cardinals_season
Content Summary:
 Momentum would be slowed by injuries, as starting pitcher Mark
    Mulder was ... Batting. edit. Note: G = Games played
    ; AB = At bats; R = Runs; H = Hits; 2B ...

3. Title: Athletics (baseball) - Wikipedia
URL: https://en.wikipedia.org/wiki/Athletics_(baseball)
Content Summary:
 ... mark when a player bats. Prior to the mid-2010s, the A
    's had a long ... Mark Mulder · P, 2000  200
... (content truncated to save space)
```

**Assistant:**
Tool Calls: query_on_page(url=https://en.wikipedia.org/wiki/Mark_Mulder, question=What is Mark Mulder's batting hand?)

**Tool Response (query_on_page):**

```
Mark Mulder's batting hand is left.
```

**Assistant:**
Tool Calls: search(query=Corey Patterson batting hand)

**Tool Response (search):**

```
1. Title: Corey Patterson - Wikipedia
```

```
URL: https://en.wikipedia.org/wiki/Corey_Patterson
Content Summary:
 Atlanta, Georgia, U.S.. Batted: Left. Threw: Right. MLB
     debut. September 18, 2000, for the Chicago Cubs.

2. Title: All-Star Final Vote - Wikipedia
URL: https://en.wikipedia.org/wiki/All-Star_Final_Vote
Content Summary:
 On Monday July 7, 2003, Kenny Lofton was added to the ballot to
     replace the Chicago Cubs' Corey Patterson who was
     injured on the day before. ... hand and wearing ...

3. Title: Cody Ross - Wikipedia
URL: https://en.wikipedia.org/wiki/Cody_Ross
Content Summary:
 He is one of the few Major League players to bat right-
     handed and throw left-handed. Cody Ross. Ross with the San
     Francisco Giants in 2011. Outfielder.
... (content truncated to save space)
```

**Assistant:** Tool Calls: query_on_page(url=https://en.wikipedia.org/wiki/Corey_Patterson, question=What is Corey Patterson's batting hand?)

**Tool Response (query_on_page):**

```
Corey Patterson's batting hand is left.
```

**Assistant:**
Tool Calls: search(query=Jeff Austin batting hand)

**Tool Response (search):**

```
1. Title: Austin Hedges - Wikipedia
URL: https://en.wikipedia.org/wiki/Austin_Hedges
Content Summary:
 Austin Charles Hedges (born August 18, 1992) is an American
     professional baseball catcher for the Cleveland
     Guardians of Major League Baseball (MLB).

2. Title: List of Major League Baseball batting champions -
     Wikipedia
URL: https://en.wikipedia.org/.../
     List_of_Major_League_Baseball_batting_champ...
Content Summary:
 In baseball, batting average (AVG) is a measure of a
     batter's success rate in achieving a hit during an at 
     bat. In Major League Baseball (MLB), ...

3. Title: Austin Riley - Wikipedia
URL: https://en.wikipedia.org/wiki/Austin_Riley
Content Summary:
 Memphis, Tennessee, U.S.. Bats: Right. Throws: Right. MLB
     debut. May 15, 2019, for the Atlanta Braves.
... (content truncated to save space)
```

**Assistant:**
Tool Calls: query_on_page(url=https://en.wikipedia.org/wiki/Jeff_Austin, question=What is Jeff Austin's batting hand?)

**Tool Response (query_on_page):**

```
The content provided does not mention Jeff Austin's batting hand.
    However, it does identify Jeff Austin as a "American Major
```

```
        League Baseball pitcher," which implies he was a baseball player
        . Pitchers typically do not bat, so the question about his
        batting hand might not be applicable. If you are referring to a
        specific Jeff Austin in the context of baseball, additional
        information would be needed to determine his batting hand.
```

**Assistant:**
Tool Calls: search(query=J.D. Drew batting hand)

**Tool Response (search):**

```
1. Title: J. D. Drew - Wikipedia
URL: https://en.wikipedia.org/wiki/J._D._Drew
Content Summary:
 On August 9, on what would have been Drew's first game
     in Philadelphia, he sat out, citing a bruised right hand. ... Drew had a very good season, batting .284&
     nbsp;...

2. Title: Home run - Wikipedia
URL: https://en.wikipedia.org/wiki/Home_run
Content Summary:
 J. D. Drew has been part of two different sets of back-to-
     back-to-back-to ... 'Ralph,' he said, 'the right-hand batters who hit to the right field are driving&nbsp
     ;...

3. Title: Adam Wainwright - Wikipedia
URL: https://en.wikipedia.org/wiki/Adam_Wainwright
Content Summary:
 After the 2003 season, the Braves traded him to the Cardinals for
     outfielder J. D. Drew. ... batting practice on
     February 21. Three days later, the ...

... (content truncated to save space)
```

**Assistant:**
Tool Calls: query_on_page(url=https://en.wikipedia.org/wiki/J._D._Drew, question=What is J. D. Drew's batting hand?)

**Tool Response (query_on_page):**

```
J. D. Drew's batting hand is left.
```

**Assistant:**
Tool Calls: search(query=Jeff Austin batting hand 1998 MLB draft)

**Tool Response (search):**

```
1. Title: 1998 Major League Baseball draft - Wikipedia
URL: https://en.wikipedia.org/wiki/1998_Major_League_Baseball_draft
Content Summary:
 The 1998 Major League Baseball draft, was the
     choosing of high school and college baseball players,
     held on June 2 and 3, 1998. A total of 1445 players were
      ...

2. Title: List of Boston Red Sox first-round draft picks -
    Wikipedia
URL: https://en.wikipedia.org/.../List_of_Boston_Red_Sox_first-
    round_draft_pic...
```

```
Content Summary:
 The Boston Red Sox are a Major League Baseball (MLB)
     franchise based in Boston, Massachusetts. They play in the
     American League East division.

3. Title: List of Detroit Tigers first-round draft picks -
    Wikipedia
URL: https://en.wikipedia.org/wiki/List_of_Detroit_Tigers_first-
    round_draft_picks
Content Summary:
 The Detroit Tigers are a Major League Baseball (MLB)
     franchise based in Detroit, Michigan. They play in the
     American League Central division.

... (content truncated to save space)
```

**Assistant:**
Tool Calls: query_on_page(url=https://en.wikipedia.org/wiki/Jeff_Austin_(baseball), question=What is Jeff Austin's batting hand?)

**Tool Response (query_on_page):**

```
Jeff Austin's batting hand is **Right**.
```

**Assistant:**
The information retrieved confirms that the batting hands of the first five picks in the 1998 MLB draft are:
1. **Pat Burrell**: Right-handed 2. **Mark Mulder**: Left-handed 3. **Corey Patterson**: Left-handed 4. **Jeff Austin**: Right-handed 5. **J. D. Drew**: Left-handed
This matches the statement provided.
Final answer: Correct.
So the answer is The batting hands of the first five picks in the 1998 MLB draft are as follows:Pat Burrell: Right-handed, Mark Mulder: Left-handed, Corey Patterson: Left-handed, Jeff Austin: Right-handed, J. D. Drew: Left-handed.

**Final Answer:**
The batting hands of the first five picks in the 1998 MLB draft are as follows:Pat Burrell: Right-handed, Mark Mulder: Left-handed, Corey Patterson: Left-handed, Jeff Austin: Right-handed, J. D. Drew: Left-handed.

## Case study 2.1 (WebSeer-14B)

**System:**
You are a reasoning assistant with the ability to perform web searches and execute Python code to help you process the content of the page and answer the user's question accurately. Do not use any knowledge you know; all facts in your thinking should be obtained from the information returned by the tools. You can repeat the search process multiple times if necessary. Once you have all the information you need, continue your reasoning. Please first make a plan before calling tools. Please answer the following question. You should provide your final answer to the "submit_answer" tool.

**User:**
Question: What German medium tank, sometimes called the "Mark V", that was deployed on the Eastern and Western fronts from mid-1943 provided the armor and suspension for the chassis of the tank destroyer built by Nazi Germany during World War II?

**Assistant:**
Tool Calls: search(query=German medium tank Mark V WWII)

**Tool Response (search):**

```
1. Title: Panther tank - Wikipedia
URL: https://en.wikipedia.org/wiki/Panther_tank
Content Summary:
```

```
  The Panther tank, officially Panzerkampfwagen V Panther (
      abbreviated Pz.Kpfw. V) with ordnance inventory
      designation: Sd.Kfz. 171, is a German medium tank&nbsp
      ;...

2. Title: Mark V tank - Wikipedia
URL: https://en.wikipedia.org/wiki/Mark_V_tank
Content Summary:
 There are eleven surviving Mark V tanks. The Mark VIII 
      tank was an enlarged Mark V with greater power: only
       those with the Liberty engine saw post-war service ...

3. Title: German tanks in World War II - Wikipedia
URL: https://en.wikipedia.org/wiki/German_tanks_in_World_War_II
Content Summary:
 List of tanks · Grosstraktor · Leichttraktor
      · Panzer I · Panzer II · Panzer III &
      middot; Panzer IV · Neubaufahrzeug ·
      Panther.

... (content truncated to save space)
```

**Assistant:**
Tool Calls: query_on_page(url=https://en.wikipedia.org/wiki/Panther_tank, question=What German medium tank, sometimes called the 'Mark V', was deployed on the Eastern and Western fronts from mid-1943?)

**Tool Response (query_on_page):**

```
The German medium tank, sometimes called the 'Mark V', that was
    deployed on the Eastern and Western fronts from mid-1943 is the
    **Panther tank**, officially designated as the **
    Panzerkampfwagen V Panther**.
```

**Assistant:**
Tool Calls: query_on_page(url=https://en.wikipedia.org/wiki/Panther_tank, question=Which tank destroyer built by Nazi Germany during World War II used the Panther tank's armor and suspension for its chassis?)

**Tool Response (query_on_page):**

```
The tank destroyer built by Nazi Germany during World War II that
    used the Panther tank's armor and suspension for its chassis was
     the **Jagdpanther**.

The Jagdpanther was a heavy Jagdpanzer (tank destroyer) that was
    based on the Panther tank's chassis, incorporating its armor and
     suspension system. It was equipped with an 88 mm long-barrel
    gun, and was designed to provide superior armor protection and
    mobility compared to other German tank destroyers of the time.
```

**Assistant:**
The German medium tank, sometimes called the "Mark V," that was deployed on the Eastern and Western fronts from mid-1943 is the Panther tank (officially Panzerkampfwagen V Panther). The tank destroyer built by Nazi Germany during World War II that used the Panther tank's armor and suspension for its chassis is the Jagdpanther.
Final answer: Panther tank.

So the answer is Panther tankWait, I need to verify the answer:
Tool Calls: search(query=German medium tank Mark V WWII)

**Tool Response (search):**

```
1. Title: Panther tank - Wikipedia
URL: https://en.wikipedia.org/wiki/Panther_tank
Content Summary:
 The Panther tank, officially Panzerkampfwagen V Panther (
     abbreviated Pz.Kpfw. V) with ordnance inventory
     designation: Sd.Kfz. 171, is a German medium tank&nbsp
     ;...

2. Title: Mark V tank - Wikipedia
URL: https://en.wikipedia.org/wiki/Mark_V_tank
Content Summary:
 There are eleven surviving Mark V tanks. The Mark VIII 
     tank was an enlarged Mark V with greater power: only
      those with the Liberty engine saw post-war service ...

3. Title: German tanks in World War II - Wikipedia
URL: https://en.wikipedia.org/wiki/German_tanks_in_World_War_II
Content Summary:
 List of tanks · Grosstraktor · Leichttraktor
     · Panzer I · Panzer II · Panzer III &
     middot; Panzer IV · Neubaufahrzeug ·
     Panther.

... (content truncated to save space)
```

**Assistant:**
Tool Calls: query_on_page(url=https://en.wikipedia.org/wiki/Panther_tank, question=Was the Panther tank deployed on the Eastern and Western fronts from mid-1943?)

**Tool Response (query_on_page):**

```
Yes, the Panther tank was deployed on both the Eastern and Western
    fronts from mid-1943. On the Eastern Front, it was introduced
    during the Battle of Kursk in the summer of 1943, and it was
    used throughout the war. On the Western Front, it was deployed
    in France, with the first Panther-equipped Panzer regiments
    arriving there in June 1944. The tank served in most European
    theatres of World War II from mid-1943 to the end of the war in
    May 1945.
```

**Assistant:**
Tool Calls: query_on_page(url=https://en.wikipedia.org/wiki/Jagdpanther, question=What tank destroyer used the Panther tank's armor and suspension for its chassis?)

**Tool Response (query_on_page):**

```
The tank destroyer that used the Panther tank's armor and
    suspension for its chassis is the **Jagdpanther**.
```

**Assistant:**
The Panther tank was indeed deployed on both the Eastern and Western fronts from mid-1943, and the tank destroyer that used its armor and suspension for its chassis is the Jagdpanther.

Final answer: Correct
So the answer is Panther tank
Tool Calls: submit_answer(answer=Panther tank)

**Final Answer:**
Panther tank

## C SMALLER MODELS BEHAVIOR BEFORE SFT

---

**Case study 3.1 (Qwen2.5-3B-Instruct):**

**System:**
You are a reasoning assistant with the ability to perform web searches and execute Python code to help you process the content of the page and answer the user's question accurately. Do not use any knowledge you know; all facts in your thinking should be obtained from the information returned by the tools. You can repeat the search process multiple times if necessary. Once you have all the information you need, continue your reasoning. Please answer the following question. You should provide your final answer to the "submit_answer" tool.

**User:**
Question: What is the profession of Ansel Elgort's character in "Baby Driver?"

**Assistant:**
Tool Calls: query_on_page(url=https://en.wikipedia.org/wiki/Baby_(film), question=What is the profession of Ansel Elgort's character in Baby Driver?)

**Tool Response (query_on_page):**

```
Ansel Elgort's character in *Baby Driver* is **a getaway driver**.
    He is a skilled driver who uses his expertise to help a group of
     robbers escape after heists, while also dealing with his
    personal challenges, including a severe hearing condition that
    requires him to wear headphones constantly. His profession as a
    driver is central to the film's plot, which revolves around high
    -speed chases and his role in the criminal underworld.
```

**Assistant:**
Tool Calls: submit_answer(answer=Ansel Elgort's character in 'Baby Driver' is a getaway driver.)

**Final Answer:**
Ansel Elgort's character in 'Baby Driver' is a getaway driver.

---

**Case study 3.2 (Qwen2.5-7B-Instruct): Smaller models typically perform short**

**System:**
You are a reasoning assistant with the ability to perform web searches and execute Python code to help you process the content of the page and answer the user's question accurately. Do not use any knowledge you know; all facts in your thinking should be obtained from the information returned by the tools. You can repeat the search process multiple times if necessary. Once you have all the information you need, continue your reasoning. Please answer the following question. You should provide your final answer to the "submit_answer" tool.

**User:**
Question: What is the profession of Ansel Elgort's character in "Baby Driver?"

**Assistant:**
Let's search for information about the movie "Baby Driver" and the character played by Ansel Elgort.
Tool Calls: search(query=Ansel Elgort character profession Baby Driver)

---

**Tool Response (search):**

```
1. Title: Baby Driver - Wikipedia
URL: https://en.wikipedia.org/wiki/Baby_Driver
Content Summary:
 Baby Driver is a 2017 action crime film written and directed by
     Edgar Wright. It stars Ansel Elgort as a getaway driver seeking
      freedom from a life of crime with his girlfriend Debora (Lily
     James). Kevin Spacey, Eiza Gonz lez , Jon Bernthal , Jon Hamm ,
     and Jamie Foxx appear in supporting roles.

2. Title: Ansel Elgort - Wikipedia
URL: https://en.wikipedia.org/wiki/Ansel_Elgort
Content Summary:
 Musical career In 2017, Elgort played the title character in Edgar
     Wright's action thriller Baby Driver , for which he received a
     Golden Globe Award nomination ...

3. Title: Ansel Elgort - IMDb
URL: https://www.imdb.com/name/nm5052065/
Content Summary:
 Ansel played the title role in Baby Driver (2017), director Edgar
     Wright's action film , starring opposite Lily James and Kevin
     Spacey. Baby Driver was ...

... (content truncated to save space)
```

---

**Assistant:**
From the search results, we can gather that Ansel Elgort plays the character Baby in the film "Baby Driver". The film describes Baby as a getaway driver who is coerced into working for a crime boss. Therefore, Baby's profession in the movie is a getaway driver.
Now I will use the query_on_page function to verify this information from the official IMDb page of the movie "Baby Driver".
Tool Calls: query_on_page(url=https://www.imdb.com/title/tt3890160/, question=What is the profession of Ansel Elgort's character in Baby Driver?)

**Tool Response (query_on_page):**

```
Failed to get page content
```

---

**Assistant:**
It seems that there was an issue retrieving the content from the IMDb page of "Baby Driver". However, based on the previous search results, we can reasonably conclude that Ansel Elgort's character, Baby, is a getaway driver in the movie.
Let's proceed with submitting this answer.
Tool Calls: submit_answer(answer=getaway driver)

**Final Answer:**
getaway driver

## D  QUALITATIVE EXAMPLE IN 7B MODELS

Case studies 3.1 and 3.2 present examples where a 7B model answered correctly before SFT but incorrectly after SFT. Before SFT, the Qwen2.5-7B model performs a single, targeted search and directly finds the answer in the tool output. After SFT, however, the model imitates the long trajectories seen in the demonstrations and issues multiple tool calls, but many of these calls query repeated or irrelevant information. The correct evidence (Marked with a red background) becomes buried in the extended context and is effectively ignored by the model. In contrast, Qwen2.5-14B-SFT

handles the same long tool-call chain and successfully identifies the correct supporting evidence, demonstrating that larger capacity is necessary for robust long-context reasoning.

---

**Case study 4.1 (Qwen2.5-7B)**

**User:**
Question: Little Brown Stein is a rivalry trophy that imitates a mug made out of what item?

**Assistant:**
Let's search for information about the Little Brown Stein trophy to find out what it is made of.
Tool Calls: search(query=Little Brown Stein trophy material)

**Tool Response (search):**

```
1. Title: Little Brown Stein - Wikipedia
URL: https://en.wikipedia.org/wiki/Little_Brown_Stein
Content Summary:
 The trophy is, as the name implies, a large stein mug with the
     results of all the games between the two painted on.

2. Title: History of FCS Rivalry Trophies - Reddit
URL: https://www.reddit.com/r/fcs/comments/1740vfs/
     history_of_fcs_rivalry_trophies/
Content Summary:
 With this week being The Battle for the Little Brown Stein, I am
     looking into the history of FCS rivalry trophies.

3. Title: 156 college football rivalry trophies, kind of ranked -
     Banner Society
URL: https://www.bannersociety.com/2019/11/28/20984367/rivalry-
     trophies-college-football-best-list-ranking
Content Summary:
 Little Brown Jug, Michigan-Minnesota, JUG, Melee weapon. Little
     Brown Stein, Idaho-Montana, STEIN, Melee weapon. Mayor's Cup,
     Rice-SMU, Silver ...

... (content truncated to save space)
```

---

**Assistant:**
From the search results, we can gather the following information:
1. The Little Brown Stein trophy is described as a large stein mug with the results of all the games between the two teams painted on it. This description matches the trophy given to the winner of the Idaho-Montana football rivalry game.
2. In the Wikipedia article, it mentions that the trophy is made of wood, but does not specify the exact type of wood used.
Given these points, it seems the Little Brown Stein trophy is likely made of wood, as mentioned in the Wikipedia article. However, without a more specific source confirming the material, we cannot be absolutely certain.
Let's verify this information using the Wikipedia page directly.
Tool Calls: query_on_page(url=https://en.wikipedia.org/wiki/Little_Brown_Stein, question=What is the material of the Little Brown Stein trophy?)

**Tool Response (query_on_page):**

```
The **Little Brown Stein** trophy is a **large stein mug**. The
     content explicitly describes it as a "large stein mug with the
     results of all the games between the two [teams] painted on."
     While the term "stein" traditionally refers to a type of ceramic
      or stoneware drinking vessel, the provided text does not
     specify the exact material beyond the term "stein mug."
```

---

```
    Therefore, based on the given information, the trophy's material
       is described as a **stein mug**.

**Answer:** The Little Brown Stein trophy is a large stein mug.
```

---

**Assistant:**
Based on the information provided by the Wikipedia page, the Little Brown Stein trophy is described as a large stein mug. While the term "stein" traditionally refers to a type of ceramic or stoneware drinking vessel, the exact material is not specified beyond the term "stein mug."
Therefore, the most accurate answer to the question is that the Little Brown Stein trophy is made out of a stein mug, which is typically made of ceramic or stoneware.
Let's submit this answer.
Tool Calls: submit_answer(answer=The Little Brown Stein trophy is made out of a stein mug, which is typically made of ceramic or stoneware.)

---

**Final Answer:**
The Little Brown Stein trophy is made out of a stein mug, which is typically made of ceramic or stoneware.

---

Case study 4.2 (Qwen2.5-7B-SFT)

**User:**
Question: Little Brown Stein is a rivalry trophy that imitates a mug made out of what item?

**Assistant:**
To answer this question, I need to determine the material used to make the Little Brown Stein rivalry trophy. This trophy is known to resemble a mug but is not made of actual ceramic. Let me perform a search to find out the specific material used for its construction.
Tool Calls: search(query=Little Brown Stein rivalry trophy material)

**Tool Response (search):**

```
1. Title: Little Brown Stein - Wikipedia
URL: https://en.wikipedia.org/wiki/Little_Brown_Stein
Content Summary:
 The trophy is, as the name implies, a large stein mug with the
     results of all the games between the two painted on.

2. Title: The 25 Best Rivalry Trophies in College Football : r/CFB
     - Reddit
URL: https://www.reddit.com/r/CFB/comments/1ccvrps/
     the_25_best_rivalry_trophies_in_college_football/
Content Summary:
 ... trophy. The Little Brown Stein (very different story from the
     Little Brown Jug) and the Dakota Marker are awesome. Upvote 7.
     Downvote Award

3. Title: The 10 best rivalry trophies in college football |
     DraftKings Network
URL: https://dknetwork.draftkings.com/2023/11/24/the-10-best-
     rivalry-trophies-in-college-football/
Content Summary:
 The trophy is a corn replica made of bronze, because of course it
     is. ... rivalry series, and the Grizzlies took home the Brown
     Stein. The ...

... (content truncated to save space)
```

---

**Assistant:**

Useful Information Summary: 1. The Little Brown Stein is a rivalry trophy between Idaho and Montana universities. 2. The trophy resembles a large stein mug. 3. The trophy was crafted by a local carpenter. 4. It costs $25 to make. 5. The trophy has been awarded since 1938.
Thinking Process: Based on the search results, particularly the information from the page at URL "https://en.wikipedia.org/wiki/Little_Brown_Stein" and other related articles, it is clear that the Little Brown Stein trophy is made of wood. The fact that it was crafted by a local carpenter and described as resembling a large stein mug indicates that it is a wooden replica of a mug.
Final Answer: Wooden.
So the answer is Wooden.Wait, I need to verify the answer before submitting it.
Tool Calls: search(query=Little Brown Stein trophy material)

**Tool Response (search):**

```
1. Title: Little Brown Stein - Wikipedia
URL: https://en.wikipedia.org/wiki/Little_Brown_Stein
Content Summary:
 The trophy is, as the name implies, a large stein mug with the
     results of all the games between the two painted on.

2. Title: History of FCS Rivalry Trophies - Reddit
URL: https://www.reddit.com/r/fcs/comments/1740vfs/
     history_of_fcs_rivalry_trophies/
Content Summary:
 With this week being The Battle for the Little Brown Stein, I am
     looking into the history of FCS rivalry trophies.

3. Title: 156 college football rivalry trophies, kind of ranked -
     Banner Society
URL: https://www.bannersociety.com/2019/11/28/20984367/rivalry-
     trophies-college-football-best-list-ranking
Content Summary:
 Little Brown Jug, Michigan-Minnesota, JUG, Melee weapon. Little
     Brown Stein, Idaho-Montana, STEIN, Melee weapon. Mayor's Cup,
     Rice-SMU, Silver ...

... (content truncated to save space)
```

**Assistant:**
The useful information from the search results includes:
1. The Little Brown Stein is a rivalry trophy between Idaho and Montana universities. 2. It is described as a large stein mug with game results painted on it. 3. The trophy has been awarded since 1938. 4. It is associated with the rivalry between Idaho and Montana football teams.
Now, let's analyze the question and the proposed answer:
**Question:** Little Brown Stein is a rivalry trophy that imitates a mug made out of what item?
**Proposed Answer:** Wooden
From the information gathered, we know that the Little Brown Stein is a large stein mug-like trophy. However, there is no direct mention in the summaries provided about the material used to make the trophy. The descriptions focus on its appearance and history rather than its composition.
Since the question specifically asks about the material used to make the trophy, and none of the summaries explicitly state the material, we cannot confirm if the trophy is made of wood or another substance. Therefore, the current answer "Wooden" is speculative and lacks concrete evidence.
To verify the accuracy of the answer, I would need to conduct a more detailed search focusing on the specific material used for the Little Brown Stein trophy. I will perform an additional search to find this information.
Tool Calls: search(query=Little Brown Stein trophy material)

**Tool Response (search):**

```
1. Title: Little Brown Stein - Wikipedia
URL: https://en.wikipedia.org/wiki/Little_Brown_Stein
Content Summary:
```

```
  The trophy is, as the name implies, a large stein mug with the
      results of all the games between the two painted on.

2. Title: History of FCS Rivalry Trophies - Reddit
URL: https://www.reddit.com/r/fcs/comments/1740vfs/
    history_of_fcs_rivalry_trophies/
Content Summary:
 With this week being The Battle for the Little Brown Stein, I am
      looking into the history of FCS rivalry trophies.

3. Title: 156 college football rivalry trophies, kind of ranked -
    Banner Society
URL: https://www.bannersociety.com/2019/11/28/20984367/rivalry-
    trophies-college-football-best-list-ranking
Content Summary:
 Little Brown Jug, Michigan-Minnesota, JUG, Melee weapon. Little
    Brown Stein, Idaho-Montana, STEIN, Melee weapon. Mayor's Cup,
    Rice-SMU, Silver ...

... (content truncated to save space)
```

**Assistant:**
None of the search results provide explicit information about the material used to make the Little Brown Stein trophy. The descriptions focus on its appearance as a large stein mug and its historical context, but they do not mention the specific material.

To verify whether the trophy is made of wood, I need to look for additional details about the trophy's construction. I will conduct a more focused search specifically targeting the material used for the Little Brown Stein trophy.

Tool Calls: search(query=Little Brown Stein trophy material made of)

**Tool Response (search):**

```
1. Title: Little Brown Stein - Wikipedia
URL: https://en.wikipedia.org/wiki/Little_Brown_Stein
Content Summary:
 Missing: material made

2. Title: Beer stein - Wikipedia
URL: https://en.wikipedia.org/wiki/Beer_stein
Content Summary:
```
` Such steins may be made out of stoneware, pewter, porcelain or even silver,`
```
wood or crystal glass; they may have open tops or hinged pewter
    lids with a thumb- ...

3. Title: History of FCS Rivalry Trophies - Reddit
URL: https://www.reddit.com/r/fcs/comments/1740vfs/
    history_of_fcs_rivalry_trophies/
Content Summary:
 With this week being The Battle for the Little Brown Stein, I am
      looking into the history of FCS rivalry trophies.

... (content truncated to save space)
```

**Assistant:**
None of the search results directly mention the material used to make the Little Brown Stein trophy. While some pages discuss beer steins and trophies in general, none specifically address the composition of the Little Brown Stein.

Given this lack of direct information, I cannot confirm whether the trophy is made of wood or another material based solely on the data retrieved so far. To provide an accurate answer, I would need to locate additional sources that specifically detail the construction materials of the Little Brown Stein trophy. Since the search results did not yield the required information, I cannot submit an answer at this time. I will need to perform further research or consult authoritative sources about the trophy's construction to determine the correct material used for the Little Brown Stein.

I will attempt one more search focusing on the Little Brown Stein trophy itself to see if any details about its material are available.

Tool Calls: search(query=Little Brown Stein trophy made of)

**Tool Response (search):**

```
1. Title: Griz set to battle Vandals for Little Brown Stein -
    Montana Athletics
URL: https://gogriz.com/news/2023/10/10/football-griz-set-to-battle
    -vandals-for-little-brown-stein
Content Summary:
 Montana and Idaho first squared off 120 years ago in 1903 and meet
     on the gridiron for the 89th time this week, with the winner
     taking home "The Little Brown Stein," a wooden traveling trophy
     . The Stein was created by UM student newspaper writer John T.

2. Title: Little Brown Stein - Wikipedia
URL: https://en.wikipedia.org/wiki/Little_Brown_Stein
Content Summary:
 The trophy is, as the name implies, a large stein mug with the
     results of all the games between the two painted on.

3. Title: History of FCS Rivalry Trophies - Reddit
URL: https://www.reddit.com/r/fcs/comments/1740vfs/
    history_of_fcs_rivalry_trophies/
Content Summary:
 With this week being The Battle for the Little Brown Stein, I am
     looking into the history of FCS rivalry trophies.

... (content truncated to save space)
```

---

**Assistant:**

Useful Information Summary: 1. The Little Brown Stein is a wooden traveling trophy between the University of Idaho and the University of Montana football teams. 2. It is described as a large stein mug with the results of their games painted on it. 3. The trophy was created by UM student newspaper writer John T. Campbell in 1938. 4. It is a unique trophy with a dark stain, but the specific type of wood is not mentioned in the summaries.

Thinking Process: From the search results, we learn that the Little Brown Stein is a wooden trophy, specifically described as a large stein mug with the results of the games painted on it. The trophy was crafted in 1938 by a UM student newspaper writer named John T. Campbell. Although the exact type of wood is not specified in the summaries, the presence of the term "wooden" in multiple contexts strongly suggests that the trophy is indeed made of wood.

Based on this analysis, the answer "Wooden" appears to be correct.

Final Answer: Wooden.

So the answer is Wooden

Tool Calls: submit_answer(answer=Wooden)

**Final Answer:**

Wooden

# E   FAILURE PATTERN ON SMALL MODELS

In Case study 5.1, the Qwen2.5-7B model did not properly close JSON tags when outputting tool calls. In Case study 5.2, the Qwen2.5-7B model kept repeatedly calling tools after a search failure until it exceeded the context length.

---

**Case study 5.1 (Qwen2.5-7B-RL)**

**System:**

```
You are a reasoning assistant with the ability to perform web
    searches and execute Python code to help you process the content
     of the page and answer the user's question accurately.
Do not use any knowledge you know; all facts in your thinking
    should be obtained from the information returned by the tools.
    You can repeat the search process multiple times if necessary.
Once you have all the information you need, continue your reasoning
    .
Please answer the following question. You should provide your final
     answer to the "submit_answer" tool. /no_think
For each function call, return a json object with function name and
    arguments within <tool_call></tool_call> XML tags:
<tool_call>
{"name": <function-name>, "arguments": <args-json-object>}
</tool_call>
```

**Prompt:**

```
Question: What is the the "Billboard" 200 where "Vice Verses"
    debuted at number 8?
```

**Assistant:**

```
To answer the question regarding the Billboard 200 chart debut of "
    Vice Verses," I will first identify the song "Vice Verses" and
    determine its chart performance. Then, I will confirm if it
    debuted at #8 on the Billboard 200. Let's proceed with this step
    -by-step.
<tool_call>
{"name": "search", "arguments": "query": "Billboard 200 chart debut
     of Vice Verses"}}
</tool_call>
```

---

**Case study 5.2 (Qwen2.5-7B-RL)**

**User:**

**Assistant:**

```
To answer this question, I will perform a web search to find
    information about the Hoover Company's sale in 2006 and the
    number of employees of the holding company involved in the sale.
<tool_call>
{"name": "search", "arguments": {"query": "The Hoover Company 2006
    sale to holding company number of employees"}}
</tool_call>
```

**tool_response:**

```
<tool_response>
1. Title: Maytag - Wikipedia
URL: https://en.wikipedia.org/wiki/Maytag
Content:
 Number of employees. 2,500. Parent ·
     Whirlpool Corporation (2006-present) ... On April
```

```
     1, 2006, Whirlpool completed its acquisition of Maytag
    Corporation.
2. Title: Whirlpool Corporation - Wikipedia
URL: https://en.wikipedia.org/wiki/Whirlpool_Corporation
Content:
 In 2023, the Fortune 500 company had an annual revenue of
    approximately $19 billion in sales, around 59,000 
    employees, and more than 55 manufacturing and ...
3. Title: Ford Motor Company - Wikipedia
URL: https://en.wikipedia.org/wiki/Ford_Motor_Company
Content:
 Ford's former British subsidiaries Jaguar and Land Rover,
    acquired in 1989 and 2000, respectively, were sold to
    the Indian automaker Tata Motors in March 2008.
4. Title: UnitedHealth Group - Wikipedia
URL: https://en.wikipedia.org/wiki/UnitedHealth_Group
Content:
 Number of employees. c. 400,000 (2024). Subsidiaries &
    middot; Optum ... In February 2006, the company
    acquired John Deere Health Care.
5. Title: The Hoover Company - Wikipedia
URL: https://en.wikipedia.org/wiki/The_Hoover_Company
Content:
 ... sales in many parts of Europe. Today, the
    Hoover Europe brand ... The machine was so fondly
    remembered that it was reintroduced and sold from 2006 to 2009.
Please summarize the useful information in the returned results
    first, then explain your thinking. The above information will
    disappear soon.
</tool_response>
```

**Assistant:**

```
Based on the information retrieved:
1. The Hoover Company's sale details are not explicitly detailed on
    its Wikipedia page.
2. There's no direct confirmation of any holding company that
    purchased Hoover in 2006.
3. However, the sale of the Hoover Company (or its European brands)
    is referenced in the context of its sales in Europe and its re-
    introduction, but again, no specific buyer is named or linked to
    the sale event.
The most relevant link is to the "Hoover Company" Wikipedia page.
    It does not explicitly mention a sale in 2006 with a buyer and
    their employment number. Other pages like Whirlpool's mention
    acquisitions but not in connection to Hoover specifically. There
    's an indirect mention that Whirlpool Corporation (acquiring
    another company in 2006) was mentioned but unrelated to Hoover.
Since no relevant data is found, and there's confusion or lack of
    clarity, I will perform a more detailed search about the Hoover
    Company.
<tool_call>
{"name": "search", "arguments": {"query": "Hoover Company sale 2006
    acquisition details"}}
</tool_call>
```

## F  MULTI-SEED EVALUATION

We report the performance of SRRL and GRPO across multiple random seeds to assess the stability and variance of both methods. The detailed results are shown in Table 4.

Table 4: Performance across different random seeds for SRRL and GRPO.

| Seed | SRRL | GRPO |
|------|------|------|
| 42 | 70.9 | 66.3 |
| 142 | 71.5 | 67.5 |
| 1142 | 69.6 | 68.1 |
| **Mean** | 70.7 | 67.3 |
| **Std** | 0.97 | 0.91 |

# G  ABLATION STUDIES ON REWARD DESIGN

To quantify the contribution of individual reward components, we conduct ablations targeting (i) the formatting penalty $R_{\text{format}}$ and (ii) the exponential discount coefficient $\alpha$ in the correctness reward $R_{\text{correct}}$. The results highlight the delicate interplay between structural constraints and incentives for iterative refinement.

## G.1  EFFECT OF THE FORMATTING PENALTY $R_{\text{FORMAT}}$

As shown in Figure 6, the formatting constraint is essential for maintaining stable training dynamics. Removing $R_{\text{format}}$ leads to a characteristic *length-explosion* failure mode: although early-stage performance (within the first 50 optimization steps) matches the full reward configuration, the policy quickly discovers a reward-hacking strategy. Without penalties on trajectory structure, the model aggressively accumulates tool calls and intermediate outputs to marginally improve the probability of producing a correct answer and thereby secure $R_{\text{correct}}$. Reinforcement learning amplifies this brittle behavior, creating a positive feedback loop in which sequence length grows without bound. Once trajectories exceed the 32k-token context window, optimization collapses entirely. These findings demonstrate that structural constraints are not auxiliary but instead *prevent pathological reward-seeking behaviors* that undermine training stability.

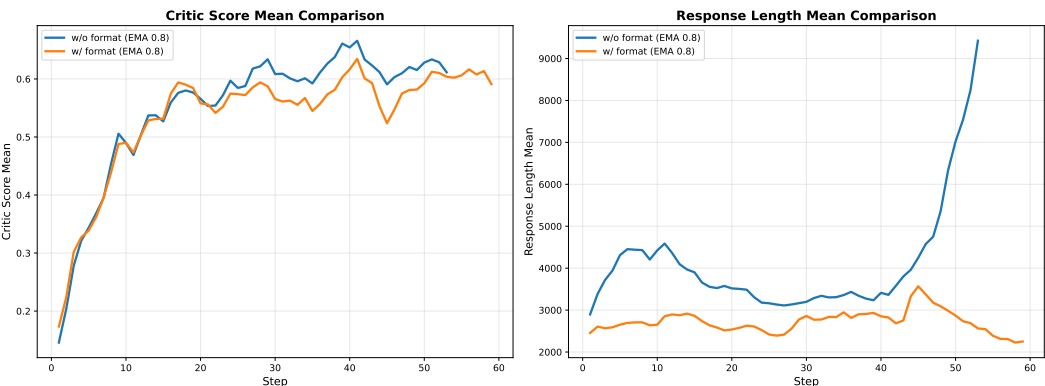

Figure 6: Accuracy and changes in output length before and after removing $R_{\text{format}}$

## G.2  EFFECT OF THE DISCOUNT COEFFICIENT $\alpha$ IN $R_{\text{CORRECT}}$

We further study the sensitivity of the correctness reward to the exponential discount factor $\alpha$, which modulates the influence of repeated attempts. As summarized in Table 5, the choice of $\alpha$ induces a clear trade-off between encouraging self-correction and avoiding inefficient exploration. When $\alpha = 1.0$ (no discount), the agent over-relies on environment feedback and tends to spam submissions, resulting in reduced accuracy despite shorter trajectories. Conversely, a steep discount ($\alpha = 0.4$) discourages refinement: after an initial failure, the sharply diminished return of additional attempts causes the model to terminate prematurely. The intermediate value $\alpha = 0.8$ achieves the best balance, yielding the highest accuracy while maintaining reasonable trajectory lengths. Overall, these results indicate that carefully tuned discounting is crucial for enabling deliberate self-reflection while discouraging indefinite, low-confidence guessing strategies.

Table 5: Performance across values of the discount coefficient $\alpha$.

| Metric | $\alpha = 0.4$ | $\alpha = 0.6$ | $\alpha = 0.8$ | $\alpha = 1.0$ |
|---|---|---|---|---|
| Accuracy | 65.2 | 69.2 | 70.9 | 68.8 |
| Length | 4524 | 3807 | 3253 | 2301 |

## H  THE USAGE OF LLM

Portions of this paper were polished by the large language models (LLMs), which were used to improve the clarity, grammar, and presentation of the text. The models were not used to generate research ideas, conduct experiments, or analyze results; all conceptual contributions and empirical findings are the work of the authors. We carefully reviewed and edited all LLM-generated suggestions to ensure accuracy and alignment with the intended meaning.

## I  THE USAGE OF LLM

Portions of this paper were polished by the large language models (LLMs), which were used to improve the clarity, grammar, and presentation of the text. The models were not used to generate research ideas, conduct experiments, or analyze results; all conceptual contributions and empirical findings are the work of the authors. We carefully reviewed and edited all LLM-generated suggestions to ensure accuracy and alignment with the intended meaning.

