# OpenReview forum: "WebSeer: Training Deeper Search Agents through Reinforcement Learning with Self-Reflection"
_ICLR.cc/2026/Conference — ICLR 2026 Poster_

### Official Review · Reviewer_NZPh · 2025-10-29

**Soundness:** 2
**Presentation:** 3
**Contribution:** 2
**Rating:** 4
**Confidence:** 4

**Summary:**

The paper proposes WebSeer, a search agent capable of performing multi-turn, real-world searches for QA tasks. The authors introduce a two-stage training pipeline that includes SFT and RL. After training, WebSeer demonstrates improvements in both the number of effective tool calls and accuracy.

**Strengths:**

1. Training an autonomous search agent capable of conducting deep, strategic, and reflective searches addresses an important and challenging research problem, and also has significant practical application value.

2. WebSeer extends to real-world search scenarios and incorporates more reasonable tools, such as Webpage Reader, enhancing its applicability.

3. The method achieves SOTA performance on HotPotQA and SimpleQA benchmarks.

**Weaknesses:**

1. In the experiments (Table 1), WebSeer (a 14B model) is compared against baselines using smaller 7B/8B models. This raises uncertainty about whether the observed improvements result from the proposed method itself or simply from scaling up model size.

2. The SRRL method is a key contribution of the paper. However, Figure 5 (top) shows that SRRL only improves accuracy by 3.6% on HotPotQA and 2.9% on SimpleQA. Were the experiments conducted multiple times to assess variance? The relatively small gains make it unclear whether the improvement is statistically significant.

3. A main feature of SRRL is that it allows the model to submit multiple answers within a single dialogue turn. However, there is no ablation study isolating the effect of this design choice. How would performance change under a conventional setting that allows only one submission during training?

4. At Line 241, the paper claims that its RL framework unifies SFT and RL under a self-reflection mechanism. However, Sec 2.4 does not describe how this unification is achieved. Based on the content, SRRL appears to directly apply GRPO for policy optimization.

5. (minor) Lines 182-183 contains repeated sentences.

**Questions:**

1. The paper emphasizes that WebSeer benefits from a self-reflection mechanism. But in Case study 1.2, the agent's outputs show no reasoning or reflection step. It directly generate tool calls in each round. Could the author clarify how self-reflection is implemented?

---

> ### Author Response · Authors · 2025-11-27
> **Rebuttal (1/3)**
>
> ## **Response to W1: Clarification on Model Size and Method Effectiveness**
>
> We would like to clarify that we focus on the 14B model is not to create an unfair comparison, but is dirven by a critical experimental finding regarding **model capacity and task complexity**, which we discussed in Section 3.3 (See Table 3).
> To demonstrate the strong effectiveness of our approach, we selected as many relevant baselines as possible from existing methods, which naturally include models of different scales such as 7B/8B and 14B. What we intend to highlight is that our model not only outperforms smaller-scale baselines but also surpasses strong 14B baselines (e.g., Qwen2.5-14B, Search-r1-14B). This setup allows for a fair and comprehensive comparison across scales.
>
> As shown in our ablation study (Table 3 in the manuscript), we conducted rigorous experiments across 3B, 7B, and 14B scales. We observed a distinct trend. For 7B scale, equipping with our SRRL on SFT stage actually degrades performance (e.g., HotpotQA -5.86%, SimpleQA -1.17% compared to the base Instruct model). The 7B model struggles to balance the complex tool-use instructions with reasoning, leading to confusion (increased tool calls but lower accuracy). For 14B scale, equipping with our SRRL on SFT stage successfully absorbs a self-reflection mechanism, yielding significant gains (HotpotQA +5.86%, SimpleQA +10.74%).
> To further investigate the relationship between model scale and our method's effectiveness, we conducted additional experiments on **Qwen2.5-32B** and **Qwen2.5-72B**. They effectively leverage our SSRL method to extend tool-use chains and improve answer accuracy.
>
> | Size | 7B | 14B | 32B | 72B |
> |------|------------|-----------|------------|-----------|
> |Qwen2.5-Instruct   | 51.95      | 62.89      | 64.84 | 68.90 |
> | WebSeer (SFT) |46.09     | 68.75     |70.72 | 72.30|
>
> ---
>
> ## **Response to W2: Clarification on Performance Interpretation and Robustness**
> We would like to clarify the interpretation of the results reported in Figure 5. The improvements shown in Figure 5 (top) represent the gain of applying our SRRL method compared to directly applying GRPO, rather than gains over the base model itself. A improvement of 3.6% (HotpotQA) and 2.9% (SimpleQA) indicates that SRRL addresses limitations in GRPO by leveraging self-reflection mechanism.
>
> | Method                       | HotpotQA Acc | HotpotQA Tool Call | SimpleQA Acc | SimpleQA Tool Call |
> |-----------------------------|--------------|----------------------|--------------|----------------------|
> | **Base**               |   62.89      |  3.57            |   65.43     |        3.76        |
> | *w/ GRPO*           | 67.27        | 7.38                | 75.98        | 6.15                |
> | *w/ SRRL* (WebSeer)  | 70.90        | 7.91                | 78.91        | 8.61                |
>
>
>
> To assess the full impact of our proposed framework, we present a performance comparison between the baselines and our WebSeer model below. The results indicate that WebSeer consistently outperforms the baselines by a significant margin, which provides a robust solution for complex reasoning tasks, validating the overall efficacy of our framework.
> | **Method** | **Inference Environment** | **NQ** | **TQ** | **Hotpot** | **2Wiki** | **In Avg** | **Musique** | **Bamb** | **PopQA** | **Out Avg** |
> |-----------|----------------------------|-------|-------|-----------|-----------|------------|-------------|----------|----------|-------------|
> | **14B Models** |||||||||||
> | Qwen2.5-14B w/ Tools | Local RAG | 72.1 | 83.8 | 62.9 | 70.9 | 72.4 | 29.7 | 72.0 | 46.1 | 49.3 |
> | **WebSeer** | Local RAG | 81.9 | 86.7 | 70.9 | 76.0 | 78.9 | 35.0 | **81.6** | **60.6** | **59.1** |
> | --- |  |  |
> | Qwen2.5-14B w/ Tools | Web Search | 72.5 | 87.9 | 67.9 | 80.3 | 77.2 | 26.6 | 73.6 | 54.7 | 51.6 |
> | **WebSeer** | Web Search | **82.8** | **91.0** | **72.3** | **84.2** | **82.6** | 35.2 | 80.0 | 58.0 | 57.7 |
>
> Besides, we have conducted variance analysis to validate these findings. We repeated the experiments 3 times with different random seeds.  Our SRRL achieved an average F1 of 70.7 (std: 0.97) on HotpotQA, compared to the score 67.3 (std: 0.91) of baseline (GRPO). Given that the performance margin (~3.4%) significantly exceeds the standard deviations, we confirm that the improvement is robust and consistent, rather than a result of random fluctuation.
> | Seed | SRRL              | GRPO              |
> |------|-------------------|-------------------|
> | 42   | 70.9              | 66.3              |
> | 142  | 71.5              | 67.5              |
> | 1142 | 69.6              | 68.1              |
> | mean | 70.7 (std = 0.97) | 67.3 (std = 0.91) |

---

> ### Author Response · Authors · 2025-11-27
> **Rebuttal (2/3)**
>
> ## **Response to W3: Ablation Study on Multiple Submissions**
> We respectfully highlight that the ablation study isolating the effect of multiple submissions was already included in our original manuscript, demonstrated in Figure 5 (top).
>
> In our experiments, the row labeled **"w/ GRPO"** represents the setting where the model is restricted to a **single submission**. The result shows that a single-submission setting achieves an accuracy of **67.27%** on HotpotQA. In contrast, enabling the multiple-submission mechanism (SRRL) boosts performance to **70.90%**. The improvements (**+3.63%** on HotpotQA and **+2.93%** on SimpleQA) demonstrate that the performance gains are directly attributed to the multi-turn reflection capability provided by SRRL.
>
> | Method                     | Acc (HotpotQA)  |  Acc (SimpleQA) |
> |---------------------------|----------------   |------------------------|
> | SRRL(multi-submission setting) | 70.90               | 78.91                 |
> | Single-submission setting     | 67.27               | 75.98                  |
> | $\Delta$                  | +3.63              | +2.93                  |
>
>
> ---
>
> ## **Response to W4: Difference from Standard GRPO**
> We apologize for the confusion caused by the description in Section 2.4. We have revised the manuscript to clarify that the "unification" refers to the **integration of behavioral learning (SFT) and outcome optimization (RL) within the self-reflection loop.**
>
> While standard GRPO focuses on **single-turn policy optimization**, SRRL unifies SFT and RL to construct a robust **Self-Reflective Agent**. SRRL bridges the gap between "mimicking reflection" (SFT) and "successful reasoning" (RL). Unlike standard pipelines where RL simply aligns preferences within a single-turn setting, our framework uses RL to explicitly reinforce the **iterative correction loop** learned in SFT, effectively unifying them into a cohesive reasoning agent.
>
> **Difference from Standard GRPO**:
> Based on this unified view, SRRL extends the application of GRPO from single-pass generation to a **multi-turn interactive paradigm**:
>
> 1. **Paradigm Shift: Interactive Self-Correction**:
>
> Standard GRPO typically optimizes a policy for single-pass correctness—expecting the model to generate the correct answer in one go. In contrast, SRRL fundamentally alters the interaction dynamics by unifying SFT and RL under a self-reflection mechanism. It treats the reasoning process as a multi-turn interaction where the model can submit answers, receive textual feedback, and refine its trajectory. Consequently, SRRL optimizes the model not just to "answer correctly," but to detect errors and recover through iterative reflection.
>
> 2. **Specialized Reward Design**:
>
> To support this multi-turn paradigm without encouraging "guessing," SRRL employs a custom trajectory-wise reward function.
>
> - Efficiency Discount: We introduce an exponential discount factor $\alpha$ to the correctness reward ($R_{correct} = r \cdot \alpha^T$), where $T$ is the number of attempts. This explicitly penalizes reliance on brute-force retries and encourages the model to reflect efficiently.
>
> - Format Constraints: We also incorporate a format reward ($R_{format}$) that imposes soft and hard penalties on output length and structure, preventing the "collapse" into repetition or malformed tool calls often seen in standard RL for long-context tasks.
>
> ---
>
> ## **Response to W5: Minor Typos**
> We thank the reviewer for pointing out the typos and minor editorial issues. We have carefully reviewed and revised the entire manuscript to correct all identified mistakes.

---

> ### Author Response · Authors · 2025-11-27
> **Rebuttal (3/3)**
>
> ## **Response to Question 1: Case Studies for Tool Calls.**
> We clarify that Self-Reflection in WebSeer manifests in two forms: Explicit (verbal) and Implicit (procedural), depending on the complexity of the task.
>
> **1. Implicit Reflection**：
> In Case Study 1.2, the agent exhibits **Procedural Reflection**. The "reflection" is embedded in the iterative refinement of actions rather than verbose monologue. When the agent observes unsatisfactory tool outputs, it implicitly reflects by generating a different query or switching tools in the next step (e.g., search $\to$ query_on_page $\to$ re-search). This represents a Procedural Reflection loop: Action $\to$ Observe $\to$ Correct.
>
> This conciseness is an **emergent behavior optimized by SRRL**. As noted in recent research (Yao et al., 2025), excessive Chain-of-Thought can sometimes lead to hallucinations or redundancy. Since our SRRL stage rewards efficiency alongside accuracy, the model learns to suppress "surface-level reasoning" (e.g., "I should try another search...") for straightforward tasks like Case 1.2, directly executing the corrective action instead.
>
> **2. Explicit Reflection**：
> Explicit verbal reasoning is retained for complex verification scenarios. In Case Study 2.1, where the evidence is ambiguous, the model triggers a verbal verification loop before the final submission: “Wait, I need to verify the answer: Tool Calls: search(…), query_on_page(…), …” followed by a final submit_answer.
>
> [1] Yao Z, Liu Y, Chen Y, et al. Are Reasoning Models More Prone to Hallucination?[J]. arXiv preprint arXiv:2505.23646, 2025.

---

### Official Review · Reviewer_g4kW · 2025-10-31

**Soundness:** 3
**Presentation:** 2
**Contribution:** 3
**Rating:** 4
**Confidence:** 3

**Summary:**

This paper introduces a search agent designed for multi-step question-answering on the web. The agent's core innovation is a two-stage training framework: 1) using multi-turn rejection sampling, where a reasoner model's proposed answers are validated by a separate verifier, to generate a high-quality dataset of self-correcting reasoning paths. 2) using Self-Reflective Reinforcement Learning,  where the agent can submit multiple answers and receive textual feedback on its performance, allowing it to iteratively refine its strategy.

**Strengths:**

The research problem—improving the depth and self-correction capabilities of search agents—is interesting and important, as current agentic systems may struggle with error accumulation and shallow reasoning.


The proposed two-stage training framework, which combines supervised fine-tuning on curated reflective trajectories with reinforcement learning, is a logical approach to the problem.


The reward mechanism in the RL stage, which provides explicit feedback on answer quality and allows for multiple submission attempts, is a reasonable design choice.

**Weaknesses:**

Overall, the presentation of the methodology could be further improved.

For example, the paper's methodology relies on an independent "verifier" model (V) to generate the SFT dataset. The description of this verifier is insufficient (only mentioned between line 188 and line 197). It is unclear how the verifier is implemented. If the verifier itself is unreliable or follows an incorrect tool-use path, its judgments would be flawed, which could introduce noise or incorrect patterns into the SFT dataset and undermine the entire training pipeline. The paper needs to detail the verifier's implementation.


In addition, the term "F1-based feedback" is not standard in reinforcement learning literature. While the paper explains it as a feedback signal, it should be more formally defined as part of the observation space or reward function to avoid ambiguity. (It is possible that the authors refer to F1 score in classification. But according to line 253, it appears that F1 is calculated based on one predicted answer and one ground-truth answer, while the F1 score in classification should be calculated based on a set of predictions and GT answers.)


(Minor issue) The paper requires careful proofreading. For instance, lines 182 and 183 contain nearly identical, repetitive sentences: "To address this limitation, we propose a multi-turn rejection sampling method to collect reasoning paths that incorporate reflective patterns. To overcome this limitation, we propose a multi-turn rejection sampling method that collects reasoning paths enriched with reflective patterns."

**Questions:**

Please refer to weakness #1 and #2.

---

> ### Author Response · Authors · 2025-11-27
> **Rebuttal**
>
> ## **Response to W1: Clarification & Implementation & Reliability of the Verifier**
>
> **1. Clarification on Verifier**
>
> We thank the reviewer for pointing out the ambiguity regarding the SFT verifier. We would like to clarify a misunderstanding: **our SFT verifier is not a passive scoring model used merely to filter answers**.
>
> Instead, it is an active, generative agent designed to synthesize "self-reflection" trajectories. Its role is to validate the answer by **attempting to reconstruct the reasoning path**. Unlike standard verification where $V(Q, A) \rightarrow [0,1]$, our verifier takes the query $Q$ and the candidate answer $A$ (proposed by the Reasoner $G$) as input. It then acts as a "prover," attempting to generate a valid **exploration trajectory** or **reasoning chain** ($\tau$) that logically leads to $A$.
>
> **Crucially, this verifier is an agent equipped with tool-invocation capabilities.** The verification process is dynamic and stepwise: at each step, the agent directly invokes tools and executes actions. Only upon the successful execution of the current step does the verifier transition to the subsequent reasoning stage. If the agent's final execution result matches the candidate answer $A$, it confirms that a valid trajectory to $A$ has been successfully reconstructed. Conversely, if the agent fails to derive $A$ within a fixed computational budget, the trajectory is discarded as invalid.
>
> **2. Implementation of Verifier**
> We have added a dedicated section in the **Appendix A** with the full prompt templates and the following implementation. Specifically, we employ **Qwen3-235B-A22B** as the verfier model to generate and verify reasoning trajectories. We chose this model for its strong reasoning and tool-use capabilities. Due to we utilize the non-thinking mode of Qwen3-235B-A22B as our verifer, we follow the suggested decoding setting with **Temperature=0.7, TopP=0.8, TopK=20, and MinP=0**. For each query, we set the retry buget as $K$ = 10.
>
> **3. Reliability of Verifier**
> To evaluate whether the verifier produces logically sound trajectories rather than superficial or lucky matches, we have conducted a human evaluation on **100 randomly sampled verifier-accepted trajectories**. We focus on verifier-accepted trajectories because these are the ones included in the SFT dataset and thus directly determine the quality of supervision provided to the model. Three expert annotators examined:
> - whether retrieved evidence supported intermediate reasoning steps;
> - whethe the final answer followed from the executed tool-based reasoning chain.
> The results show strong agreement between human judgments and verifier decisions, indicating that the verifier's correctness predicate reliably selects both label-correct and reasoning-sound trajectories.
> - **Trajectories-Answer Matching** (100% Agreement): Human annotators verified that all verifier-accepted trajectories successfully reached the correct final answer.
> - **Evidence-Reasoning Alignment** (High Agreement): Annotators found that nearly all verifier-accepted trajectories exhibited coherent multi-step reasoning that was grounded in the retrieved evidence. While minor variations existed in how annotators interpreted certain intermediate steps, no cases were identified where the final answer was reached through unsupported or hallucinated reasoning.
>
> ---
>
> ## **Response to W2: Clarifying the Definition and Use of the Token-Level F1 Reward**
> Thank you for pointing out the potential ambiguity. We agree that the term “F1-based feedback” may be interpreted as the classification F1-score, which is not our intention. In our work, the “F1” refers to the **token-level overlap F1 metric** commonly used in extractive QA (e.g., SQuAD), computed between **a single predicted answer and a single reference answer**. To avoid confusion, we have revised the terminology and now explicitly describe it as “**token-level F1 reward**”.
>
> ---
>
> ## **Response to W3: Typos**
> We thank the reviewer for pointing out the typos and minor editorial issues. We have carefully reviewed and revised the entire manuscript to correct all identified mistakes.

---

### Official Review · Reviewer_KqpE · 2025-10-31

**Soundness:** 3
**Presentation:** 3
**Contribution:** 3
**Rating:** 6
**Confidence:** 2

**Summary:**

This paper tackles the challenge of training open-domain, web-based question-answering agents capable of multi-hop reasoning and robust self-correction. Existing web agents often stop prematurely, fail to verify their reasoning, or overfit to closed-corpus retrieval. The proposed system, WebSeer, aims to build deeper and more reflective reasoning trajectories by combining:

- Self-Reflective Supervised Fine-Tuning (SFT): multi-turn rejection-sampling trajectories verified by an independent verifier model, teaching the agent to backtrack and self-correct.

- Self-Reflective Reinforcement Learning (SRRL): a reinforcement phase where the model can submit intermediate answers multiple times, receive F1-based feedback, and continue improving under a shaped reward.

The final agent interacts with real web tools (search, webpage reader, code executor) and is evaluated across seven multi-hop QA benchmarks, achieving state-of-the-art performance and strong generalization to out-of-distribution datasets.

Contributions:

Introduces a two-stage self-reflection training pipeline that operationalizes “reason-verify-refine” behaviors in web agents.

Proposes a self-reflective RL framework (SRRL) that allows iterative answer submission and feedback-guided reasoning.

Designs a trajectory-level reward that balances answer correctness, brevity, and depth of reasoning.

Demonstrates substantial improvements over strong baselines (Search-r1, Qwen2.5-14B, etc.) on both in-domain and open-web QA datasets.

**Strengths:**

Clear motivation and novelty: The idea of explicit self-reflection integrated into both SFT and RL phases is well justified and fills a gap between shallow tool use and deep verification-oriented reasoning.

Strong empirical performance: WebSeer consistently outperforms baselines on seven benchmarks, with notable OOD generalization (e.g., +15.9 on NQ, +27.2 on 2Wiki, +4.0 avg over Qwen2.5-14B).

Well-structured training pipeline: The paper carefully separates data construction, SFT masking (to avoid overfitting to observations), and SRRL fine-tuning, providing a replicable framework.

Insightful analysis: The ablation studies show how self-reflective trajectories and multi-submission feedback both contribute to deeper tool usage and higher final accuracy.

**Weaknesses:**

Limited transparency of verifier reliability: The paper assumes the verifier’s correctness predicate $\psi$
 yields high-quality reflective trajectories, but quantitative verifier accuracy or error propagation analysis is missing.

Reward design justification: The choice of exponential discount on multiple submissions is somewhat ad-hoc; an analysis of sensitivity to discount parameters or exploration–exploitation balance would strengthen claims.

Computational cost: The reflective data collection (multi-turn verifier rejection sampling) seems expensive; runtime statistics or data efficiency analysis are lacking.

Ablation on real-web vs synthetic setting: While generalization results are good, it remains unclear whether the model trained with restricted search data directly scales to dynamic, noisy real-web environments.

Clarity and formalism: Some algorithmic details (e.g., GRPO vs DAPO implementation differences) could be more rigorously formalized.

**Questions:**

How accurate is the independent verifier during trajectory construction? Did you measure verifier precision/recall or its effect on SFT data quality?

How sensitive is WebSeer’s performance to the number of verifier retries
$K$ and the validity predicate threshold?

In SRRL, how did you balance the trade-off between early termination (short trajectories) and the benefit of deeper reflection?

Could the approach generalize beyond QA (e.g., web-based decision-making or scientific reasoning tasks)?

What is the relative wall-clock cost of SFT vs SRRL stages, and is RL training feasible for larger backbones (e.g., 70B models)?

Would a joint training of verifier and reasoner be possible or beneficial compared to separate optimization?

---

> ### Author Response · Authors · 2025-11-27
> **Rebuttal (1/4)**
>
> ## **Response to W1 & Q1: Clarification & Human Eval on Verifier**
> **1. Human Evaluation on Verifier**
> To evaluate whether the verifier produces logically sound trajectories rather than superficial or lucky matches, we have conducted a human evaluation on **100 randomly sampled verifier-accepted trajectories**. We focus on verifier-accepted trajectories because these are the ones included in the SFT dataset and thus directly determine the quality of supervision provided to the model. Three expert annotators examined:
>
> - whether retrieved evidence supported intermediate reasoning steps;
>
> - whether the final answer followed from the executed tool-based reasoning chain.
>
> The results show strong agreement between human judgments and verifier decisions, indicating that the verifier's correctness predicate reliably selects both label-correct and reasoning-sound trajectories.
>
> - **Trajectories-Answer Matching (100% Agreement)**: Human annotators verified that all verifier-accepted trajectories successfully reached the correct final answer.
> - **Evidence-Reasoning Alignment (High Agreement)**: Annotators found that nearly all verifier-accepted trajectories exhibited coherent multi-step reasoning that was grounded in the retrieved evidence. While minor variations existed in how annotators interpreted certain intermediate steps, no cases were identified where the final answer was reached through unsupported or hallucinated reasoning.
>
> Combined with our **ground-truth–guided rejection sampling (Section 2.3)**, which accepts a trajectory only if the verifier’s conclusion matches the dataset ground truth, the system provides a strong structural safeguard against error propagation: any verifier hallucination automatically results in rejection rather than incorrect data entering SFT.
> **2. Precision/Recall is not a suitable metric for our verifier**
> The verifier is **not a binary classifier**. It is a **constructive prover** that attempts to **generate and execute** a tool-use trajectory to validate an answer. Thus, precision/recall metrics are not directly applicable.
>
> ---
>
> ## **Response to W2: Ablation Study for Exponential Discount** $\alpha$
> We conducted a sensitivity analysis on the submission discount parameter $\alpha$, which discounts rewards based on the number of attempts. The results on HotpotQA  reveal a clear trade-off between encouraging iterative refinement and maintaining reasoning efficiency.
>
> - $\alpha = 1.0$ (no discount): Instead of exploring more, the model tends to take the shortest path. Without the incentive to be rigorous, the model may revert to "shallow tool-use" patterns. This behavior inflates chain lengths (avg. 2301 tokens) and leads to suboptimal accuracy (68.8%).
> - $\alpha = 0.4$: The reward decays rapidly with each failed submission attempt. This creates immense pressure on the agent to get the answer correct on the very first try. To ensure this, the model becomes overly cautious, performing extensive searches and reasoning (resulting in a length of 4524 tokens) to verify every detail before daring to submit. And accuracy drops to 68.8%, the lowest across settings.
> - $\alpha = 0.6$: The discount is moderate and reduces over-submission, but still suppresses deeper self-correction. After an incorrect first attempt, the agent often finds the remaining reward insufficient to justify multi-step refinement. Consequently, chain lengths remain moderate (avg. 3807 tokens) and accuracy reaches 69.2%.
> - $\alpha=0.8$ (our default) produces the strongest accuracy and the most stable chain-length distribution. The model is encouraged to perform meaningful self-reflection after a failed attempt, yet discouraged from repeatedly submitting low-confidence guesses. As a result, chain lengths remain well-controlled (3253 tokens) and accuracy peaks at 70.9%, the highest across all tested configurations.
>
> | Metric   | α = 0.4 | α = 0.6 | α = 0.8 | α = 1.0 |
> |----------|---------|---------|---------|---------|
> | Accuracy | 65.2    | 69.2    | 70.9    | 68.8    |
> | Length   | 4524    | 3807    | 3253    | 2301    |

---

> ### Author Response · Authors · 2025-11-27
> **Rebuttal (2/4)**
>
> ## **Response to W3: Cost of SFT Data Construction**
> We clarify that our data construction is highly cost-efficient.
>
> **Open-Source & Local Deployment**: All synthetic trajectories were generated using the open-source Qwen3-235B-A22B model deployed on our own infrastructure, avoiding the substantial per-token costs typically associated with distillation from proprietary APIs.
>
> **Cost Estimate**: The pipeline produced **~2,300 successful trajectories** (after filtering with a **~40% rejection rate**), consuming roughly **200M tokens**. Because we utilized local inference, the operational cost was effectively limited to GPU compute time, totaling less than **$100 USD**. This confirms that effective self-reflective agents can be trained without expensive distillation from frontier models.
>
> ---
>
> ## **Response to W4: Generalization to Real-Web**
> We thank the reviewer for highlighting the question of generalization to dynamic and noisy real-web environments. We clarify the following points:
> 1. **Our restricted environment is not synthetic; it is a controlled real-web retrieval setting.** The restricted environment is implemented via a local RAG-style search over real webpage snapshots, not synthetic or fabricated content.
> 2. **We tested the trained agent in real webpages by live web search across several benchmarks.** Table 1 (“WebSeer with Web Search”) shows that the model maintains strong performance, indicating effective transfer to real-web conditions. This setting retrieves actual, dynamic web content rather than restricted-domain snippets, demonstrating that the model’s reasoning skills transfer beyond the controlled environment.
> 3. **We acknowledge that real-web exploration remains challenging**. Operating in fully dynamic webpages introduces additional challenges: 1) anti-bot protection, 2) unpredictable DOM structures, 3) redirections and modal dialogs, and 4) partially missing or dynamically generated content. Our current training pipeline does not explicitly optimize for robustness to these rendering-layer uncertainties, which are **largely orthogonal to the reasoning skills** targeted by our method.
>
> ---
>
> ## **Response to W5: Formailization Statements**
> We have formalized the implementation of GRPO and DAPO in Section 2.4 of the revised manuscript to explicitly show the differences.
>
> ---
>
> ## **Response to Q2: Sensitivity Analysis on Retry budget**
> We constructed SFT datasets using different verifier retry budgets K \in {3,10,20} and evaluated the resulting models on HotpotQA and SimpleQA. The results show a consistent trend:
> - **Small $K$ (e.g., $K=3$)** yields noticeable improvements over the “Before SFT” baseline, but the verifier is often unable to recover from early exploration failures. This leads to shorter trajectories and limited diversity in reflective paths.
> - **Moderate $K$ (e.g., $K=10$)** provides a substantial additional gain, as the verifier can capture “recovery” trajectories on harder queries where the model identifies an incorrect intermediate step, corrects it, and successfully backtracks.
> - **Large $K$ (e.g., K$=20$)** yields only marginal improvement (e.g., 68.75→69.10 on HotpotQA), indicating diminishing returns. Problems that remain unresolved beyond $K$ typically require knowledge not available from retrieved evidence, and thus additional retries do not contribute further useful supervision.
>
> | Retry Budget K | HotpotQA | SimpleQA |
> |----------------|----------|----------|
> | Before SFT     | 62.89    | 65.43    |
> | 3              | 66.12    | 73.80    |
> | 10             | 68.75    | 76.17    |
> | 20             | 69.10    | 76.40    |

---

> ### Author Response · Authors · 2025-11-27
> **Rebuttal (3/4)**
>
> ## **Response to Q3: Balancing Early Termination vs. Deeper Reflection in SRRL**
> The exploration–exploitation trade-off in SRRL is primarily governed by our **trajectory-level reward design**, in particular the exponential discount term $\alpha^T$. Allowing multiple submission attempts ($T$) enables the agent to **explore deeper reflection paths** when the first answer is incorrect, rather than terminating prematurely. At the same time, the discount factor $α \in (0,1)$ imposes a **soft penalty** on excessive retries, preventing unbounded exploration.
> Our sensitivity analysis quantitatively validates this trade-off:
> - $\alpha = 1.0$ (no discount) leads to premature termination: accuracy drops, and trajectories become too short (2301 tokens), indicating insufficient reflection.
> - $\alpha = 0.4$ imposes an overly harsh penalty, suppressing exploration; the model fails to self-correct, resulting in both lower accuracy and unnecessary length expansion (4524 tokens).
> - $\alpha = 0.6$ improves stability but still lacks optimal balance.
> - $\alpha=0.8$ achieves the **best accuracy** and the **most stable** trajectory length, effectively encouraging deeper reflection when needed while discouraging gratuitous retries.
> | α | Accuracy (%) | Avg. Length |
> |:-------:|:------------:|:------------:|
> | 0.4     | 65.2         | 4524         |
> | 0.6     | 69.2         | 3807         |
> | **0.8 (Ours)** | **70.9** | **3253** |
> | 1.0     | 68.8         | 2301         |
>
> ---
>
> ## **Response to Q4: Generalization beyond QA**
> The core components of WebSeer are task-agnostic, depending only on:(i) the availability of external tools, and (ii) an evaluable reward signal. Following the reviewer's suggestion, we have formulated a discussion on how the SRRL  framework extends to other domains, specifically highlighting its efficiency benefits.
>
> - Code Agent: While many existing agents rely on a "Generate $\rightarrow$ Execute $\rightarrow$ Debug" loop, WebSeer’s reflection paradigm offers a distinct advantage: Pre-execution Verification. Analogous to how WebSeer cross-verifies retrieval results before synthesizing an answer, the model can be trained to "mentalize" or statically analyze the code logic against the problem description before invoking the execution tool. This serves as a proactive filter to catch logical fallacies or misalignment with the goal (e.g., verifying if a loop condition matches the intended boundary) without incurring the cost of runtime execution. In large-scale experiments or long-horizon tasks where code execution is resource-intensive or potentially risky, this "think before you run" capability significantly reduces computational overhead and prevents error propagation, distinct from the standard retroactive debugging approach.
>
> - Mathematical Reasoning: Similarly, for complex math problems, the framework can be adapted to reflect on the logical consistency of intermediate steps before performing calculations. The "Search" tool can be augmented with a "WolframAlpha" or calculator tool, where the reflection step ensures the formulated equations accurately represent the problem constraints before the solver is called.
>
> We believe that WebSee provides a generalizable blueprint for building more cost-effective and rigorous agents across these domains byunifying cold-start and RL within a self-reflection paradigm.
>
> ---
>
> ## **Response to Q5: Cost and Scaling**
> 1. **Relative Wall-Clock Cost: SFT vs. SRRL**
> As noted in Section 3.1, our SRRL training on a 14B model requires:
> - SFT: **~10 GPU hours for 4 epochs**.
> - SRRL: **~480 A800 GPU hours for 100 RL steps** with 12×8 trajectories per step and up to 30 turns.
> 2. **Feasibility for Larger Backbones (e.g., 70B)**
> The RL algorithm and the tool interface both scale cleanly to larger backbones, with the primary bottleneck being compute. Because WebSeer already achieves strong gains with moderate RL steps atop a solid SFT checkpoint, we believe the method is feasible for 70B-scale models, especially with modern RL-efficiency techniques.
> To verify it, we conducted additional scaling experiments on **Qwen2.5-32B** and **Qwen2.5-72B**. They effectively leverage our SSRL method to extend tool-use chains and improve answer accuracy.
>
> | Model Size | Before SFT | After SFT |
> |------------|------------|-----------|
> | 7B         | 51.95      | 46.09     |
> | 14B        | 62.89      | 68.75     |
> | 32B        | 64.84      | 70.72     |
> | 72B        | 68.90      | 72.30     |

---

> ### Author Response · Authors · 2025-11-27
> **Rebuttal (4/4)**
>
> ## **Response to Q6: Joint Training of Verifier and Reasoner**
> We appreciate the reviewer’s question regarding whether jointly training the verifier and the reasoner could offer benefits over separate optimization. We discuss both the **theoretical considerations** and **practical constraints** for large-model agent frameworks.
> 1. **Joint training introduces strong instability in tool-use agents**
>
>
> In our setting, the verifier is used to validate reasoning produced by the reasoner. If trained jointly, these two components would co-adapt, where the verifier drifts to match the reasoner’s mistakes or becomes exploitable, leading to instability similar to reward hacking.
>
> 2. **Verifier and reasoner serve different roles that benefit from different objectives**
>
> The reasoner must generate diverse multi-step tool chains and refine its reasoning. The verifier must be conservative, stable, and correctness-oriented. Thus, coupling their objectives collapses this separation. This “fixed evaluator, improving actor” pattern is standard in stable RL pipelines (e.g., RLHF with fixed reward models).

---

### Official Review · Reviewer_oPma · 2025-11-01

**Soundness:** 4
**Presentation:** 3
**Contribution:** 3
**Rating:** 6
**Confidence:** 3

**Summary:**

The paper introduces WebSeer, a web search agent that learns to build deeper and more reliable tool chains for multi hop questions. It first collects long reflective trajectories by rejection sampling for supervised fine tuning, then runs a self reflective RL stage where the model can resubmit an answer, read feedback, and refine its plan. The agent uses a small toolset for search, page reading, and code execution, and it treats submitting the final answer as a tool. Training is site restricted on Wikipedia, while inference runs on the open web. A 14B model beats strong baselines on seven benchmarks and also does well on FanOutQA, FRAMES, SimpleQA, and Bamboogle. Taken together, the two stage recipe turns shallow tool use into steadier and longer chains and lets one model plan, act, verify, and decide when to stop. It also shows that learning in a clean, restricted setup can carry over to the open web with strong gains.

**Strengths:**

- Clear and practical idea: learn long, reflective tool use in a clean Wiki-only setup, then run on the open web. The two-stage SFT → SRRL recipe is simple, and “submit answer” is treated as a tool, which makes stopping explicit.
- Strong results across many benchmarks and tough OOD sets, with consistent gains over strong web-agent baselines.
- Good behavior analysis and ablations that explain why it works: tool-call depth becomes more balanced, removing SRRL hurts, and the SFT data mix matters.
- Interfaces and evaluation are well specified, with concrete tool APIs and LLM-as-a-judge prompts, plus transparent training and compute details.
- Meaningful impact: the method makes web agents plan deeper and verify better, and the recipe is easy to adopt in other agent settings.

**Weaknesses:**

- The open-web transfer story is promising but the ablations are only partial. The paper already shows that limiting RL to a single submission hurts, analyzes tool-use distributions, and studies how the SFT data mix changes behavior. What’s missing are comparisons across training regimes (restricted vs mixed vs open), a sensitivity check for the page-reading/normalization stack, and systematic curves that relate accuracy to chain length under different tool budgets, search-depth caps, and submission-discount settings. A short discussion or small exploratory study on these fronts would make the restricted-to-open claim feel more complete.

- The SFT verifier is under-specified and hard to reason about. We don’t know the model family/size, the prompt template and decoding settings, or the retry budget. It’s unclear how it accepts or rejects a candidate (string match vs semantic check), how it handles aliases and partial answers, or whether it uses the same tools and constraints as the agent. We also don’t see rates: how often it disagrees and forces a re-check, what fraction of trajectories get kept, how long those trajectories are, and how much of the SFT set ends up single-step vs multi-step. Without these basics, reproducibility is limited and it’s difficult to tell how much the verifier shapes the data distribution and later RL stability.

**Questions:**

- Restricted to open transfer
Did you consider building a training set that mixes a small portion of open-web episodes with wiki episodes, then run SFT and also SRRL on the mixed set? If yes, how did accuracy, chain length, and number of submissions change as the share of open-web episodes grew? If not, what behavior would you expect and why?

- SFT verifier details
What model do you use for the verifier? What is the prompt, decoding setup, retry budget, and timeout? What rule accepts or rejects a candidate answer, and does the verifier use the same tools and constraints as the agent?

- Verifier impact
How much does verifier stringency change the SFT distribution and later RL stability? If you lower or raise the verifier bar while keeping SFT size fixed, how do single-step share, tool-call patterns, and final accuracy move?

---

> ### Author Response · Authors · 2025-11-27
> **Rebuttal (1/3)**
>
> ## **Response to Weakness 1 & Question 1: Restricted-to-Open Transfer**
> We thank the reviewer for highlighting the promise of our open-web transfer story and for suggesting a comparison across training regimes. We agree that exploring the spectrum from restricted to fully open-web training is a critical line of inquiry.
>
> **1. Behavioral Consistency and Transferability**
>
> Our core hypothesis is that the reasoning and search patterns learned in a high-quality, restricted domain are structurally transferable to the open web. To validate this, we compared **the distribution of tool calls per trajectory** between models operating in the Open-Web domain versus the Restricted domain. The results show that there are **minor variances** at the extremes, the dominant mode of interaction (3–6 calls) remains consistent across both regimes.
>
> | Environment | < 3 Calls | 3 - 6 Calls | 7 - 9 Calls | > 10 Calls |
> | :--- | :---: | :---: | :---: | :---: |
> |**Open-Web**| 23.09 | 63.19 | 4.51 | 9.21 |
> |**Restricted**| 11.91 | 77.54 | 5.90 | 4.65 |
>
> This behavioral similarity suggests that the policy learned during restricted training is robust. The restricted environment acts as a noise-reduced training ground that teaches the agent how to search, while the open web provides the breadth of information during inference. We acknowledge that mixing open-web data into training could further increase trajectory diversity and potentially improve performance on real-web domains, and we view this as a promising direction for future work.
>
> **2. Economic and Engineering Feasibility Analysis**
>
> Beyond the instability of live web environments, we provide **a concrete cost** breakdown to demonstrate why this is prohibitively expensive for academic research compared to our restricted training setup. We calculate **the minimum overhead** for a single RL training run as follows: We use a batch size of 12 prompts, with 8 sampled trajectories per prompt. Based on our statistics, the average trajectory requires 6 search calls. This results in $12 \times 8 \times 6 = 576$ **Google search calls per single update step** . And training typically requires ~200 steps to reach stability, totaling approximately **115,200 search calls** . This scale presents **two blocking issues** :
>
> - **Rate Limits** : The official Google Search API tier often limits low-tier usage to ~1,000 calls per day. At this rate, a single training run would take over 100 days to complete.
>
> - **Financial Cost** : Commercial third-party search APIs typically cost around $10 per 1,000 requests. This translates to roughly **$1,150 per single training run**.
>
> Given that robust research requires dozens of runs for hyperparameter tuning, seed variation, and ablation studies, the cumulative cost would rapidly exceed tens of thousands of dollars. **By contrast, our restricted-domain approach eliminates these marginal costs while, as shown in Section 1, maintaining behavioral fidelity to open-web search patterns.**
>
> **3.The Ideal vs. The Practical**
>
> We agree with the reviewer’s intuition: in an unconstrained setting, training an agent directly on the mixed data or open web to master optimal exploration strategies is the "Holy Grail" of search agents. It would likely yield an agent with even stronger capabilities in navigating complex, noisy real-world data. However, given current API costs and infrastructure limits, this remains a challenge for the broader community.
>
> ---

---

> ### Author Response · Authors · 2025-11-27
> **Rebuttal (2/3)**
>
> ## **Response to Weakness 1: Accuracy curves vs. chain length under different hyperparameters**
>
> To systematically analyze how accuracy varies with chain length under different behavioral constraints, we conducted additional experiments across two key hyperparameter groups:**(i) tool-call budgets and (ii) discount factors.**
>
> **1. Tool-call budgets**
> We evaluated accuracy and corresponding chain lengths under different maximum tool-call budgets (5, 10, 20). The results reveal a clear pattern as shown in pic (https://postimg.cc/XrtzRTXd):
> - Budget=5: Chain lengths become severely truncated (Avg. 1437 tokens), and accuracy drops sharply because the agent cannot complete necessary multi-step reasoning.
> - Budget=10:This setting achieves the best **accuracy–length trade-off**, with sufficient steps to verify information while preventing unnecessary expansion. Accuracy improves to 69.50%, with moderate chain lengths (avg. 1531 tokens).
> - Budget=20: Although accuracy further increases to 71.50%, the gains are marginal compared to the substantial growth in chain length (avg. 1634 tokens). The looser constraint encourages the RL policy to over-expand trajectories, reducing efficiency and causing occasional length inflation.
>
> |          | budget = 5 | budget = 10 | budget = 20 |
> |----------|------------|-------------|-------------|
> | Accuracy | 66.02      | 69.24       | 69.50       |
> | Length   | 1437       | 1531        | 1634        |
>
>
> **2. Discount factors ($\alpha$)**
> We further examined how accuracy–chain length curves behave under different values of the submission-discount parameter $\alpha \in {0.4, 0.6, 0.8, 1.0}$. The results indicate a clear trade-off between encouraging refinement and enforcing efficiency.
> - $\alpha = 1.0$ (no discount): Instead of exploring more, the model tends to take the shortest path. Without the incentive to be rigorous, the model may revert to "shallow tool-use" patterns. This behavior inflates chain lengths (avg. 2301 tokens) and leads to suboptimal accuracy (68.8%).
> - $\alpha = 0.4$: The reward decays rapidly with each failed submission attempt. This creates immense pressure on the agent to get the answer correct on the very first try. To ensure this, the model becomes overly cautious, performing extensive searches and reasoning (resulting in a length of 4524 tokens) to verify every detail before daring to submit. And accuracy drops to 68.8%, the lowest across settings.
> - $\alpha = 0.6$: The discount is moderate and reduces over-submission, but still suppresses deeper self-correction. After an incorrect first attempt, the agent often finds the remaining reward insufficient to justify multi-step refinement. Consequently, chain lengths remain moderate (avg. 3807 tokens) and accuracy reaches 69.2%.
> - $\alpha=0.8$ (our default) produces the strongest accuracy and the most stable chain-length distribution. The model is encouraged to perform meaningful self-reflection after a failed attempt, yet discouraged from repeatedly submitting low-confidence guesses. As a result, chain lengths remain well-controlled (3253 tokens) and accuracy peaks at 70.9%, the highest across all tested configurations.
>
> | Metric   | α = 0.4 | α = 0.6 | α = 0.8 | α = 1.0 |
> |----------|---------|---------|---------|---------|
> | Accuracy | 65.2    | 69.2    | 70.9    | 68.8    |
> | Length   | 4524    | 3807    | 3253    | 2301    |
>
> ---

---

> ### Author Response · Authors · 2025-11-27
> **Rebuttal (3/3)**
>
> ## **Response to W2 & Q2: The Specification of SFT Verifer**
>
> We thank the reviewer for highlighting the missing specifications of the SFT verifier. We agree that these details are essential for reproducibility and for understanding the data distribution. We have added a dedicated section in the **Appendix A** with the full prompt templates and the following specifications:
>
> **1. Model and Decoding Setup**
>
> Specifically, we employ **Qwen3-235B-A22B** as the verfier model to generate and verify reasoning trajectories. We chose this model for its strong reasoning and tool-use capabilities.
> Due to we utilize the non-thinking mode of Qwen3-235B-A22B as our verifer, we follow the suggested decoding setting with  **Temperature=0.7, TopP=0.8, TopK=20, and MinP=0**. For each query, we set the retry buget as **$K$ = 10**. Besides, we provide the **detailed prompt** used for the verifer in Appendix A.
>
> **2. The role of verifier.**
>
> We would like to clarify a misunderstanding: our SFT verifier is not a passive scoring model used merely to filter answers. Instead, our verifier takes the query $Q$ and the candidate answer $A$ (proposed by the Reasoner $G$) as input. It then acts as a "prover," attempting to generate a valid **exploration trajectory** or reasoning chain ($\tau$) and sequentially execute it to verify whether it logically leads to $A$.
>
> Consistent with the agent (e.g., the Reasoner $G$), the verifier **operates as an agent with tool-invocation capabilities, sharing the exact same tool design and retry budget.**
>
> **3. Data Statistics and Distribution**
>
> We would like to clarify that our agent does not employ a "forced re-check" loop where it iteratively corrects itself. Therefore, the "disagreement" is equivalent to the **rejection rate** during this filtering phase. Our statistics show that 63% of the generated trajectories are kept. The average trajectory length is **4,606 tokens**. The average number of tool calls per trajectory is **6.75**.
> We explicitly controlled the complexity of the SFT dataset. As analyzed in Figure 4, we experimented with different mixing ratios. Our final model used a **ratio of 1:1.5 (single-pass correct vs. multi-refinement trajectories)** to prevent the model from overfitting to either "easy" retrieval or excessive, unnecessary reasoning.
>
> ---
>
> ## **Response to Q3: Clarification on Verifer**
> We would like to clarify that our verifier functions as a binary filter based on ground-truth correctness (execution success and answer matching), rather than a continuous scoring model. Therefore, our current dataset represents the set of all successfully solved trajectories.

---

### Official Review · Reviewer_dqp4 · 2025-11-01

**Soundness:** 3
**Presentation:** 3
**Contribution:** 3
**Rating:** 8
**Confidence:** 4

**Summary:**

This paper proposes WebSeer, a tool-use agent trained with reinforcement learning (RL) that incorporates a self-reflection mechanism. Specifically, the agent is equipped with searching, webpage reading, and code execution tools. The training process contains a supervised fine-tuning (SFT) stage followed by an RL stage. This RL stage mostly follows GRPO, but is designed to encourage self-reflection. The resulting 14B model achieves state-of-the-art results on several multi-hop question answering datasets and demonstrates strong generalization to out-of-domain datasets.

**Strengths:**

1. The paper tackles the relevant and interesting problem of training tool-use agents. It presents a solid approach based on RL, with a notable mechanism for encouraging self-reflection (though some technical details remain unclear, as noted in the questions below).
2. The empirical performance is strong. The model outperforms baselines on most target datasets and also demonstrates good OOD generalization.
3. The analysis in Sec. 3.3 provides helpful insights on behaviors of tool-use agents trained with RL and empirical lessons on how to train them.

**Weaknesses:**

1. Limited discussion of related work: There is a significant body of work on using RL for general tool use, not limited to search [1,2,3]. The paper would be stronger if it more comprehensively discussed and positioned itself relative to this broader line of related work.
2. Narrow experimental scope: The evaluation is focused on multi-hop question answering. While I think this is acceptable and does not undermine the paper's core contributions, showing the framework's effectiveness on other types of tasks would make the work even more impactful. If time and resources do not permit, it would be great if the authors can include some discussions.

[1] ReTool: Reinforcement Learning for Strategic Tool Use in LLMs
[2] Tool-Star: Empowering LLM-Brained Multi-Tool Reasoner via Reinforcement Learning
[3] Synthetic Data Generation & Multi-Step RL for Reasoning & Tool Use

**Questions:**

Questions regarding the RL algorithm:
1. In SRRL, if a rollout achieves a low reward in an earlier step but a better reward in a later step (via self-reflection), how is the advantage calculated? Is it based only on the final reward, or does it consider every reward achieved throughout the trajectory?
2. Alternatively, is every answer submission considered a separate rollout? (e.g., separate trajectories sharing initial steps but with different ending steps/submission times).
3. Following 2., if not, does it mean the model will generate a trajectory containing multiple answer submissions? In that case, how do you determine when to end an trajecotry during inference?
4. Could you elaborate on how the model's behavior differs between SRRL and regular GRPO? More analysis or examples here would be helpful.

Questions regarding experiments:
1. What is the specific base model used for training (e.g., Qwen2.5-14B or Qwen3-14B)?
2. It appears the "Qwen3-14B w/ Web Search tools" baseline is missing from Table 1. Can you add the results?
3. Figure 5 is difficult to interpret. It might be clearer to present a direct numerical comparison between your model vs. the no-SFT baseline, perhaps in a format similar to Table 5.

---

> ### Author Response · Authors · 2025-11-27
> **Rebuttal (1/2)**
>
> ## **Response to W1: Limited discussion of related work on general RL-for-tool-use**
>
> We sincerely thank the reviewer for highlighting these relevant works. We agree that positioning WebSeer within the broader context of general RL for tool use—beyond just search agents—strengthens the paper's contribution.
>
> In the revision, we have expanded the Related Work section to comprehensively discuss frameworks such as ReTool and Tool-Star. We have explicitly clarified the distinction between these generalist frameworks and WebSeer:
>
> - **Differentiation on Paradigm**: While methods like Tool-Star primarily focus on hierarchical reward design to optimize forward planning or collaborative execution, they often treat tool use as a linear or branching generation task.
> - **WebSeer's Unique Contribution**: We emphasize that WebSeer distinguishes itself by unifying explicit and implicit self-reflection. Through the SRRL paradigm, our model not only learns to explicitly generate reasoning steps to revise queries but also implicitly internalizes error-correction intuitions directly into its policy. This enables spontaneous backtracking and dynamic recovery in open-ended web environments, a capability that extends beyond the scope of prior general tool-use optimization.
>
> ---
>
> ## **Response to W2: Narrow experimental scope**
> We thank the reviewer for the positive assessment and the insightful suggestion regarding the broader applicability of our framework. We add a "Future work" section before the conclusion section:
>
> **1. Rationale for Multi-hop QA Focus:**
> We focused on multi-hop QA because it (i) provides clear ground truth and (ii) stresses long-horizon retrieval-reasoning—exactly what SRRL targets. In the revision, we will add a discussion detailing: (a) how the same SRRL interface (search/reader/code + submit-answer) and termination policy (via the Submit-Answer tool with a T_max cap) generalize to other tasks (web navigation, planning, code-based tools), and (b) design guidance for extending to those domains. As demonstrated by our SOTA results on HotpotQA and SimpleQA, this setting effectively validates the robustness of our self-reflective mechanism.
>
> **2. Discussion on Broader Applicability:**
> Following the reviewer's suggestion, we have formulated a discussion on how the SRRL  framework extends to other domains, specifically highlighting its efficiency benefits.
>
> - Code Agent: While many existing agents rely on a "Generate $\rightarrow$ Execute $\rightarrow$ Debug" loop, WebSeer’s reflection paradigm offers a distinct advantage: Pre-execution Verification. Analogous to how WebSeer cross-verifies retrieval results before synthesizing an answer, the model can be trained to "mentalize" or statically analyze the code logic against the problem description before invoking the execution tool. This serves as a proactive filter to catch logical fallacies or misalignment with the goal (e.g., verifying if a loop condition matches the intended boundary) without incurring the cost of runtime execution. In large-scale experiments or long-horizon tasks where code execution is resource-intensive or potentially risky, this "think before you run" capability significantly reduces computational overhead and prevents error propagation, distinct from the standard retroactive debugging approach.
>
> - Mathematical Reasoning: Similarly, for complex math problems, the framework can be adapted to reflect on the logical consistency of intermediate steps before performing calculations. The "Search" tool can be augmented with a "WolframAlpha" or calculator tool, where the reflection step ensures the formulated equations accurately represent the problem constraints before the solver is called.
>
> We believe that WebSeer provides a generalizable blueprint for building more cost-effective and rigorous agents across these domains by unifying cold-start and RL within a self-reflection paradigm.
>
> ---

---

> ### Author Response · Authors · 2025-11-27
> **Rebuttal (2/2)**
>
> ## **Response to Q1. Advantage calculation when earlier steps have low reward but later steps succeed**
>
> In SRRL, the policy advantage is computed from a trajectory-level final reward: the final F1 of the last submission, discounted by the number of submissions ($\alpha^T$), plus a format penalty. Intermediate r(t) are returned as text and appended to the context to drive self-reflection, but are not used for credit assignment.
> Response to Q2. Are multiple answer submissions treated as separate rollouts?
> No. A single trajectory may include multiple answer submissions because WebSeer is allowed to self-reflect and refine its reasoning chain. These submissions belong to one trajectory, and only the final submission determines the reward used for policy optimization.
>
> ---
>
> ## **Response to Q3. How does the model know when to end a trajectory during inference?**
>
> During inference, the model terminates the trajectory autonomously by invoking the Submit-Answer tool. The model is trained to use this tool only when it has gathered sufficient information to answer the user's query confidently. Unlike the training phase (where the environment might prompt for a retry upon a wrong answer), in the standard inference setting, the first Submit-Answer action typically concludes the generation.
>
> ---
>
> ## **Response to Q4. How does SRRL differ from GRPO?**
> The key differences between SRRL and GRPO lie in the interaction paradigm and the reward structure:
>
> **1. Paradigm Shift: Interactive Self-Correction:**
>
> Standard GRPO typically optimizes a policy for **single-pass correctness**—expecting the model to generate the correct answer in one go. In contrast, SRRL fundamentally alters the interaction dynamics by unifying SFT and RL under a **self-reflection mechanism**. It treats the reasoning process as a **multi-turn interaction** where the model can submit answers, receive textual feedback, and refine its trajectory. Consequently, SRRL optimizes the model not just to "answer correctly," but to detect errors and recover through iterative reflection.
>
> **2. Specialized Reward Design:**
>
> To support this multi-turn paradigm without encouraging "guessing," SRRL employs a custom trajectory-wise reward function.
>
> - Efficiency Discount: We introduce an exponential discount factor $\alpha$ to the correctness reward ($R_{correct} = r \cdot \alpha^T$), where $T$ is the number of attempts. This explicitly penalizes reliance on brute-force retries and encourages the model to reflect efficiently.
>
> - Format Constraints: We also incorporate a format reward ($R_{format}$) that imposes soft and hard penalties on output length and structure, preventing the "collapse" into repetition or malformed tool calls often seen in standard RL for long-context tasks.
>
> ---
>
> ## **Response to Q5 & Q6 &7: Experimental Details and Additional Results**
>
> **Base model used for training (Q5)**
>
> In our experiments, we adopt Qwen2.5-14B-Instruct as the backbone model.
>
> **Adding Qwen3-14B w/ Web Search to Table 1 (Q6).**
>
> We have added the results for **Qwen3-14B equipped with Web Search tools** to Table 1 in the revised manuscript. Its accuracy is 68.6% (In-Domain Avg) and 48.8% (Out-of-Domain Avg). This represents an improvement of +3.2% and +4.6% respectively over the Local RAG setting for the same model, confirming that WebSeer's benefits generalize across model versions.
>
> **Comparison with the No-SFT baseline (Q7).**
>
> We have converted Figure 5 (Bottom) into a direct numerical format similar to Table 5 to clearly contrast our full model against the No-SFT (RL-only) baseline. While WebSeer achieves high accuracy, the w/o SFT variant suffers from "training collapse". Without the structural guidance provided by SFT data, the model fails to maintain valid tool-invocation formats during RL exploration, leading to a failure in optimizing the objective.
>
> | Method                      | HotpotQA Acc | HotpotQA Tool Call | SimpleQA Acc | SimpleQA Tool Call |
> |-----------------------------|--------------|----------------------|--------------|---------------------|
> | **SRRL w/o SFT**            | 0.00         | N/A*                 | 0.00         | N/A*                |
> | **SRRL w/ SFT (WebSeer)**   | 70.90        | 7.91                 | 78.91        | 8.61                |
>
> *The model generates malformed output, making valid tool calls impossible.

---

### Official Review · Reviewer_cCUs · 2025-11-01

**Soundness:** 3
**Presentation:** 3
**Contribution:** 3
**Rating:** 6
**Confidence:** 3

**Summary:**

This paper presents WebSeer, a novel approach for training LLM search agents, i.e. agents able to perform iterative calls to Web search engines to answer a given query. The approach consists of two main stages: 1) supervised fine-tuning (SFT) on high quality multiturn search interactions,  and 2) a novel self-reflective reinforcement learning (SRRL) pipeline. The approach is used to finetuning Qwen 14B and achieves SOTA benchmark scores for it's size when given access to Web search.

**Strengths:**

- The paper studies a relevant problem: Tool calling (such as Web searches) is a very timely and challenging area of research, unveiling a lot of potential for LLMs.
- The presented WebSteer approach is sound and, to the best of my knowledge, novel.
- The experiments ablate two key choices made: 1) self-reflection and 2) the cold-start initialization via SFT, both proved to be beneficial.
- Trained LLMs reach SOTA benchmark scores for their sizes.

**Weaknesses:**

- The analysis claims that the method's effectiveness is heavily dependent on model capacity. Only the 14B model showed consistent performance and stable behavior improvements, while smaller 3B and 7B models showed degraded performance after SFT and instability during RL. I find this quite puzzling and would like to know if the authors have further explanations on why this is the case. In particular, some qualitative example after SFT or during RL would further illustrate the cause.

- The ablations focus primarily on the training methodology (SRRL, cold-start). It would be beneficial to see further ablations, e.g., on the impact of the reward design components ($R_{format}$ and $R_{correct}$).

**Questions:**

See weaknesses.

---

> ### Author Response · Authors · 2025-11-27
> **Rebuttal (1/2)**
>
> We sincerely thank the reviewer for their constructive feedback and for recognizing the potential of our work. We particularly appreciate the insightful questions regarding the underlying **causes of performance gaps across model scales** and the suggestion to conduct **further ablations on the reward design**. These comments have guided us to perform deeper analyses that significantly strengthen our paper.
>
> ---
>
> ## **Response to Weakness 1: Model Capacity Dependency**
> As noted in **Table 3** of our manuscript, SFT increases tool usage across all model scales, however, it degrades accuracy for smaller models (3B and 7B), whereas the 14B model achieves consistent improvements. Based on our further qualitative analysis, we attribute this gap to two primary failure modes inherent to smaller models in agentic tasks:
>
> **1. Context Overwhelm & Attention Dilution**
>
> The Before-SFT and After-SFT results across various model scales demonstrate that smaller models struggle to maintain reasoning coherence over long-horizon trajectories.
> | Model Size | Training Stage | Total Length | Tool Call | Accuracy        |
> |-----------|----------------|--------------|-----------|-----------------|
> | **3B**    | Before SFT     | 1774         | 4.31      | 44.73           |
> |  **3B**    | After SFT      | **8309**         | **12.40**     | **41.21 (↓3.52)**   |
> | **7B**    | Before SFT     | 1567         | 2.95      | 51.95           |
> |  **7B**    | After SFT      | **6525**         | **9.23**      | **46.09(↓5.86)**   |
> | **14B**   | Before SFT     | 1888         | 3.57      | 62.89           |
> |  **14B**   | After SFT      | 5617         | 13.43     | 68.75 (↑5.86)   |
>
> - Before SFT: Smaller models typically perform short, single-step searches and often succeed on simple queries because the context remains short. As shown in **Appendix C** of the revised manuscript, the 3B, 7B, and 14B models all exhibit this short-horizon, single-step search behavior, with smaller models relying on especially shallow tool-use trajectories.
> - After SFT: Although the models successfully learn the **pattern** of deep search (e.g., invoking multiple tools), they fail to preserve the **reasoning precision** needed to filter and prioritize the resulting larger context. As shown in **Appendix D** of the revised manuscript, the 7B model **mimics the long-chain behavior** of the SFT data but **generates redundant queries**. The resulting massive accumulation of search snippets dilutes the model's attention, causing it to "lose" the correct evidence amidst the noise. In contrast, as shown in Appendix B, the 14B model effectively attends to relevant, distinct pieces of evidence across long contexts.
>
> **2. Structural Collapse during RL**
>
> Robust instruction following is a prerequisite for RL optimization in tool-use scenarios. Smaller models exhibit high instability in maintaining structured outputs (JSON) under the pressure of RL updates. As mentioned in Section 3.3, **smaller models often suffer from instability**, generating repetitive text or malformed JSON.
> As shown in the revised manuscript, the 14B model maintains valid syntax while optimizing reasoning paths (**See Appendix B**), whereas the 7B model frequently degenerates into generating malformed JSON or repetitive text loops (**See Appendix E**). This structural collapse prevents the environment from executing tools, yielding zero rewards and effectively causing the policy to diverge.
>
> **3. Extended Scaling Experiments**
>
> To further validate that the performance degradation is strictly a result of limited model capacity, we conducted additional experiments with larger Qwen2.5 variants (32B and 72B). The results show that our method exhibits **a clear scaling law behavior**. While the 3B/7B models suffer from the "context overwhelm" described above, all models with size $\ge$ 14B consistently benefit from the SFT stage. The performance gains stabilize as model size increases, confirming that our long-horizon reflective data requires a critical capacity threshold (around 14B) to be effective.
> | Size | HotpotQA Acc | HotpotQA Tool Call | SimpleQA Acc | SimpleQA Tool Call |
> |-------|--------------|---------------------|---------------|----------------------|
> | 7B    | 51.95        | 2.95                | 46.09         | 9.23                 |
> | 14B   | 62.89        | 3.57                | 68.75         | 13.43                |
> | 32B   | 64.84        | 3.46                | 70.72         | 15.42                |
> | 72B   | 68.90        | 4.01                | 72.30         | 12.82                |

---

> ### Author Response · Authors · 2025-11-27
> **Rebuttal (2/2)**
>
> ## **Response to Weakness 2: Ablation Studies on Reward Design**
> To better understand the contribution of individual reward components, we conducted ablation studies examining the effect of removing the formatting penalty $R_{\text{format}}$ and the influence of the exponential discount $\alpha$ in $R_{\text{correct}}$.
>
> **Experiment 1: Impact of $R_{format}$**
>
> Following the required formatting pattern is critical for training stability. As visualized in https://postimg.cc/rdhtdJTr , the model exhibits length explosion when $R_{\text{format}}$ is removed. Although the initial performance (within the first 50 steps) remains comparable to the full setup, RL optimization soon triggers a **reward-hacking phenomenon (as visualized in https://postimg.cc/H8MJh1yM )**: the model discovers that exhaustive information gathering via excessive tool calls marginally increases the probability of answering correctly (thereby securing $R_{\text{correct}}$).Without the constraint of $R_{format}$, the RL algorithm reinforces this inefficient strategy, creating a vicious cycle where trajectories grow indefinitely until they exceed the context window ($32k$ tokens), leading to training collapse.
>
> **Experiment 2: Impact of $\alpha$ in $R_{\text{correct}}$**
>
> We have performed sensitivity analysis on the weighting coefficient $\alpha$ in Equation 5, which discounts rewards based on the number of attempts. The results indicate **a clear trade-off** between encouraging refinement and enforcing efficiency. With $\alpha=1.0$ (no discount), the model is under-penalized and tends to "spam" submissions, relying on environmental feedback rather than internal verification, which leads to suboptimal accuracy. Conversely, a low $\alpha$ (e.g., 0.4) imposes an overly steep penalty that discourages the model from utilizing the self-correction mechanism; if the first attempt fails, the diminished expected reward for subsequent attempts causes the model to terminate reasoning prematurely. Our chosen value of $\alpha=0.8$ strikes **the optimal balance**, effectively incentivizing the model to perform necessary self-reflections while discouraging indefinite, low-confidence guessing strategies.
> | $\alpha$ Value | HotpotQA (Acc %) | SimpleQA (Acc %) |
> | :---: | :---: | :---: |
> | 1.0 | 68.8 | 89.1 |
> | **0.8 (Ours)** | **70.9** | **90.0** |
> | 0.6 | 69.2 | 88.5 |
> | 0.4 | 65.2 | 83.1 |

---

### Author Response · Authors · 2025-12-02
**General response**

We sincerely thank the reviewers for their **dedicated time and constructive feedback**. We are greatly encouraged that the reviewers recognize the **value of our work**, highlighting our "**clear and practical idea**" with "**meaningful impact**" (Reviewer oPma) and describing the method as "**sound and... novel**" (Reviewer cCUs). Reviewers further commended the "**solid approach**" (Reviewer dqp4) and the "**well-structured training pipeline**" (Reviewer KqpE) that operationalizes reflective reasoning. We summarize the primary concerns raised across all reviews and the concrete steps we took to address them:

1.  **Verifier Specification**: Reviewers g4kW and oPma requested **deeper technical details** regarding the **verifier's implementation** and its **reliability** in generating SFT trajectories.
2.  **Model Scaling & Fairness**: Reviewers cCUs and NZPh questioned the **fairness of training on 14B model** and inquired about the **stability** of the method on smaller architectures.
3.  **Ablations**: Reviewers dqp4, cCUs, and NZPh suggested **expanding ablations** (e.g., reward design, single-submission baselines).
4.  **SRRL Mechanics**: Reviewers dqp4 and NZPh asked for the **explicit distinction between SRRL and standard GRPO**, and the specific **impact of the multi-submission design**.

**Our Response:**

1.  **Verifier Transparency**: We clarified that the verifier acts as a "**constructive prover**" designed to **generate valid reasoning chains** rather than merely scoring answers. We have added the **full implementation details and prompt templates in Appendix A**. To prove reliability, we conducted a **human evaluation** demonstrating **High consistency in Evidence-Reasoning Alignment** to validate SFT data quality.
2.  **Scalability Validation**: We **extended experiments to Qwen2.5-32B and 72B models**, confirming that our method's **gains scale effectively** and provided **variance analysis (Mean 70.7, std=0.97)** to confirm the improvements are **statistically robust**.
3.  **Ablations**: We **systematically validated** our design through **comprehensive ablations** on reward components and tool-call constraints, confirming their **critical role in balancing reasoning depth with efficiency**.
4.  **Formalization**: We clarified that unlike GRPO's single-pass optimization, SRRL introduces a paradigm shift to **Interactive Self-Correction**, **unifying SFT and RL** under a **multi-turn reflection loop**. We demonstrated that this mechanism **significantly outperforms the single-turn GRPO baseline(+3.63% on HotpotQA)**.

**All revisions have been integrated into the revised manuscript.**

---

### Meta-Review · Area_Chair_WsMp · 2026-01-04

**Summary:**

This paper introduces WebSeer, an intelligent search agent trained via reinforcement learning enhanced with a self-reflection mechanism to overcome limitations in shallow tool use and error accumulation. By training on a large reflection patterns annotated dataset with a two-stage framework, WebSeer generates longer, more effective tool-use trajectories and improves answer accuracy. A single 14B model achieves state-of-the-art performance on HotpotQA and SimpleQA and generalizes well to out-of-distribution tasks.

The strengths include: 1) the investigated problem is timely and challenging; 2) the proposed method/idea is sound and novel; 3) strong experimental results with good ablations and generalization; 4) transparent training and compute details; 5) Insightful analysis. However, the reviewer concerns include:
1. The method's effectiveness is heavily dependent on model capacity (cCUs, NZPh);
2. Missing further ablations, e.g. on the impact of the reward design components (cCUs), restricted vs open setup (oPma, KqpE), the effect of SRRL (NZPh), etc;
3. Limited discussion of related work (dqp4);
4. Missing evaluation beyond multi-hop question answering (dqp4);
5. Missing details on the SFT verifier (oPma, KqpE, g4kW);
6. Sensitivity analysis on hyperparameters (oPma, KqpE);
7. Computational cost on the reflective data collection (KqpE);
8. Clarity, presentation and formalism (KqpE, g4kW, NZPh);
9. The small gain by SRRL (NZPh).

Most of the concerns are well addressed by the rebuttal, except #4, according to the AC's point of view. For #4, the rebuttal only added some discussion in the future works, instead of providing results on other benchmarks. Although this is not ideal, it is not a major blocker for this paper to be accepted, given most of other concerns were already well addressed. Thus, the AC recommends to accept this paper, and the authors should reflect the reviewers' feedback in the final version.

**Reviewer Concerns:**

The reviewer concerns include:
1. The method's effectiveness is heavily dependent on model capacity (cCUs, NZPh);
2. Missing further ablations, e.g. on the impact of the reward design components (cCUs), restricted vs open setup (oPma, KqpE), the effect of SRRL (NZPh), etc;
3. Limited discussion of related work (dqp4);
4. Missing evaluation beyond multi-hop question answering (dqp4);
5. Missing details on the SFT verifier (oPma, KqpE, g4kW);
6. Sensitivity analysis on hyperparameters (oPma, KqpE);
7. Computational cost on the reflective data collection (KqpE);
8. Clarity, presentation and formalism (KqpE, g4kW, NZPh);
9. The small gain by SRRL (NZPh).

Most of the concerns are well addressed by the rebuttal, except #4, according to the AC's point of view. For #4, the rebuttal only added some discussion in the future works, instead of providing results on other benchmarks. Although this is not ideal, it is not a major blocker for this paper to be accepted, given most of other concerns were already well addressed.

**Reviewer Scores:**

The original scores are 6 (cCUs), 8 (dqp4), 6 (oPma), 6 (KqpE), 4 (g4kW), 4 (NZPh). Given most of the concerns are well addressed by the rebuttal, the AC thinks most of the reviewers will have positive scores on this paper.

---

### Decision · Program_Chairs · 2026-01-26

Accept (Poster)